# Lung injury-induced activated endothelial cell states persist in aging-associated progressive fibrosis

Ahmed A. Raslan [1,2,3], Tho X. Pham[1,2], Jisu Lee[1], Konstantinos Kontodimas[2,4], Andrew Tilston-Lunel [2,4], Jillian Schmottlach[1], Jeongmin Hong [1,2], Taha Dinc [1], Andreea M. Bujor [1], Nunzia Caporarello[5], Aude Thiriot [6], Ulrich H. von Andrian [6,7], Steven K. Huang[8], Roberto F. Nicosia[9], Maria Trojanowska [1,2], Xaralabos Varelas [2,4] ✉ & Giovanni Ligresti [1,2] ✉

Progressive lung fibrosis is associated with poorly understood aging-related endothelial cell dysfunction. To gain insight into endothelial cell alterations in lung fibrosis we performed single cell RNA-sequencing of bleomycin-injured lungs from young and aged mice. Analysis reveals activated cell states enriched for hypoxia, glycolysis and YAP/TAZ activity in ACKR1+ venous and TrkB+ capillary endothelial cells. Endothelial cell activation is prevalent in lungs of aged mice and can also be detected in human fibrotic lungs. Longitudinal single cell RNA-sequencing combined with lineage tracing demonstrate that endothelial activation resolves in young mouse lungs but persists in aged ones, indicating a failure of the aged vasculature to return to quiescence. Genes associated with activated lung endothelial cells states in vivo can be induced in vitro by activating YAP/TAZ. YAP/TAZ also cooperate with BDNF, a TrkB ligand that is reduced in fibrotic lungs, to promote capillary morphogenesis. These findings offer insights into aging-related lung endothelial cell dysfunction that may contribute to defective lung injury repair and persistent fibrosis.

The regenerative capacity of the lung in response to injury deteriorates with aging, contributing to disrepair and fibrosis[1–3]. Pulmonary fibrosis, including idiopathic pulmonary fibrosis (IPF), is often associated with aging and is characterized by alveolar damage and the accumulation of collagen-producing fibroblasts that contribute to excessive extracellular matrix deposition and loss of organ function[4–6]. IPF occurs largely in elderly adults, thus, age has emerged as a key risk factor for the fibrotic process[3,7]. However, mechanisms linking aging to the development of pulmonary fibrosis are poorly understood.

Studies that we and other have performed using the bleomycin model of lung fibrosis have shown that aged mice are more susceptible to develop sustained fibrogenesis compared to young animals[8–12]. Aging-associated transcriptional abnormalities in the lung endothelium have been proposed to contribute to the persistent activation of collagen-producing fibroblasts during IPF progression[1,9,13,14]. In this regard, endothelial cells (ECs) from aged mouse lungs with sustained fibrosis exhibit reduced expression of endothelial identity genes and increased expression of inflammatory- and fibrosis-associated genes

[1]Arthritis and Autoimmune Diseases Center, Department of Medicine, Boston University Chobanian and Avedisian School of Medicine, Boston, MA, USA. [2]Pulmonary Center, Department of Medicine, Boston University Chobanian and Avedisian School of Medicine, Boston, MA, USA. [3]Department of Zoology, Faculty of Science, Assiut University, Assiut, Egypt. [4]Department of Biochemistry and Cell Biology, Boston University Chobanian and Avedisian School of Medicine, Boston, MA, USA. [5]Department of Medicine, Loyola University Chicago, Chicago, IL, USA. [6]Department of Immunology, Harvard Medical School, Boston, MA, USA. [7]The Ragon Institute of MGH, MIT and Harvard, Cambridge, MA, USA. [8]Department of Internal Medicine, University of Michigan Medical School, Ann Arbor, MI, USA. [9]Department of Laboratory Medicine and Pathology, University of Washington, Seattle, WA, USA. ✉e-mail: xvarelas@bu.edu; ligresti@bu.edu

compared to young animals[8,9]. Intriguingly, loss of ERG, a transcription factor with key roles in controlling endothelial cell identity and inflammation[15,16], recapitulated transcriptional and phenotypic alterations observed in aged lungs post bleomycin injury, including aberrant EC identity and sustained fibrosis[8]. These findings suggest that aging-associated dysfunctional transcriptional mechanisms governing endothelial cell identity contribute to maladaptive injury responses and sustained fibrosis post lung injury.

Endothelial regeneration following lung injury was recently reported to be orchestrated by lung capillary endothelial progenitor cells known as general capillary (gCap) ECs[17]. gCap ECs have been suggested to behave as stem/progenitor cells, giving rise to specialized alveolar capillary ECs, known as aerocytes (aCap ECs) which are critical for gas exchange[17–19] and are often lost in human IPF lungs[20–22]. In addition to ECs from the lung capillary bed, ECs from other lung vascular beds, such those derived from veins, have been recently shown to participate in aberrant vascular remodeling associated with IPF pathogenesis[21–23]. Indeed, single cell RNA-sequencing (scRNA-seq) studies in human IPF lungs have shown that systemic venous EC greatly expand in IPF lungs and are associated with lung areas affected by aberrant collagen deposition[24].

To gain preliminary insights into unexplored features of endothelial repair and pathogenic vascular remodeling, we have carried out scRNA-seq on young and aged mouse lungs injured by a single dose of bleomycin, focusing primarily on EC dynamics that are associated with persistent lung fibrosis in aged animals. We discovered that lung injury was associated with the appearance of multiple subpopulations of activated ECs exhibiting distinct gene expression signatures that reflected their endothelial origin (capillary- and venous-derived ECs). Activated ECs derived from venous and capillary beds were the most represented EC subtypes exhibiting aberrant responses to injury with aging. Longitudinal single-cell analysis together with lineage tracing revealed that injury-associated endothelial states transiently appeared during the peak of collagen production and vanished during the resolution phase in young lungs. Conversely, ECs derived from aged lungs persisted in these activated states and were topologically restricted to fibrotic areas, indicating a failure of the aged vascular ECs to return to quiescence. Differential transcriptional analysis together with downstream pathway evaluation identified putative genes and signaling pathways associated with metabolism, such as glycolysis and oxidative phosphorylation, as well as upstream regulators, including MYC, mTOR, and YAP/TAZ that likely contribute to the appearance and pathogenic persistence of activated ECs with aging. Long-term tissue alterations, including tissue hypoxia, were observed in fibrotic aged lungs following lung injury, suggesting that transcriptional and metabolic maladaptation to injury with aging may impact the capacity of activated ECs to return to quiescence and restore normal lung homeostasis. Finally, we identified a convergent axis between YAP and TrkB signaling as a putative regulatory node with an important function in injury-induced capillary ECs activation in vivo and lung capillary morphogenesis in vitro. Our observations offer a distinctive picture of the vascular endothelial cell dynamics associated with sustained fibrosis in aged lungs and suggest that maladaptation to injury due to aging may interfere with lung ECs, thereby affecting fibrosis resolution and restoration of normal lung functions.

## Results

### Aging-associated lung vascular EC injury dynamics mapped by scRNA-seq

To define aging-associated cellular and transcriptional responses during lung injury-induced fibrosis, we employed scRNA-seq and profiled whole lungs from young mice (2 months) and aged mice (18 months) following intratracheal bleomycin instillation. Lungs were isolated at the early resolution phase of fibrosis (30 days post-bleomycin) and compared to saline-treated young and aged lungs (Supplementary Fig. 1A). After quality filtering, we obtained approximately 52,542 cell profiles, from all the samples. We then performed dimensionality reduction with canonical correlation analysis (CCA) subspace alignment followed by unsupervised clustering. We identified 15 distinct lung cell types that were defined using canonical lineage-defining markers to annotate clusters (Supplementary Fig. 1B, C and Supplementary Fig. 2), with vascular ECs being the most represented cellular subtype (nearly 50%).

To specifically investigate endothelial cell heterogeneity and transcriptional responses to injury with aging, we re-clustered lung ECs (*Cdh5+* and *Pecam-1+*) using unsupervised clustering analysis. To annotate different endothelial subpopulations, we used lineage-defining markers based on prior studies and identified 9 vascular ECs subclusters, including those from veins, arteries, general capillaries (gCap) and aerocytes (aCap) (Fig. 1A and Supplementary Fig. 3). Among these were EC subclusters that emerged exclusively in bleomycin-treated lungs, which we defined as "activated" gCap ECs, "activated" aCap ECs, "activated" arterial ECs, and "activated" venous ECs. Although these newly emerged activated EC subtypes exhibited distinct gene expression signatures defining their endothelial origin, they also shared common marker genes of cell activation, including *Fxdy5, Spp1, Ankrd37, Lrg1,* and *Amd1* which distinguished them from their quiescent counterparts (Fig. 1B). Genes associated with glycolysis were distinctly expressed in all lung EC subtypes post injury (Fig. 1C). Ingenuity Pathway Analysis (IPA) identified multiple signaling pathways that were enriched across all activated lung EC subtypes, and among them, mTOR, HIF1α, glycolysis, and Toll-like receptor signaling pathways were most representative (Fig. 1D). In addition, several transcriptional regulators were implicated as potential mediators of the gene expression changes associated with injury-mediated EC activation, including transcriptional regulators previously implicated in cell differentiation and metabolic control such as YAP1, MYC, and β-catenin (CTNNB1)[25–27] (Fig. 1E). To investigate the magnitude of EC activation in young and aged lungs following injury, we performed additional cell composition and transcriptional analysis and found that the number of ECs expressing high levels of activated marker genes was much greater in injured aged lungs compared to injured young lungs (Fig. 1F and Supplementary Fig. 4), suggesting that EC activation in aged lungs is more sustained. Genes associated with glycolysis and hypoxia, including *Hif1a* and *Bnip3*, also exhibited augmented expression in activated aged lung ECs compared to young ones (Fig. 1G), yet suggesting an incapacity of aged lung ECs to return to a normal metabolic state.

To investigate the long-term consequences of the maladaptive capillary EC metabolic responses associated with aging, we measured tissue hypoxia in young and aged mouse lungs following bleomycin injury (60 days post-injury) and correlated it with the overall capillary density within each examined area. Pimonidazole was used to detect lung hypoxia as previously described[28]. Our findings revealed that at 60 days post-injury, aged lungs, compared to young lungs, exhibited elevated levels of hypoxia which largely localized in areas with reduced capillary density (Fig. 1H). Additional immunofluorescence analysis showed that hypoxic signal was mainly detected in PDPN+ type I epithelial cells (ATI) as well as in vimentin positive (VIM+) mesenchymal cells (Supplementary Fig. 5). These findings suggest that sustained lung hypoxia and reduced capillary density in aged lungs are due to maladaptive lung EC injury responses that contribute to the metabolic dysregulation of neighboring cells. Additionally, the limited level of EC activation in young lungs relative to aged lungs during the early resolution phase of lung fibrosis (Fig. 1F) suggests that lung EC activation is reversible in young lungs.

To specifically investigate the reversibility of lung EC activation following bleomycin challenge, we carried out a parallel scRNA-seq analysis on bleomycin-injured lungs in young mice during the peak of fibrosis (day 14 post bleomycin) and early resolution phase

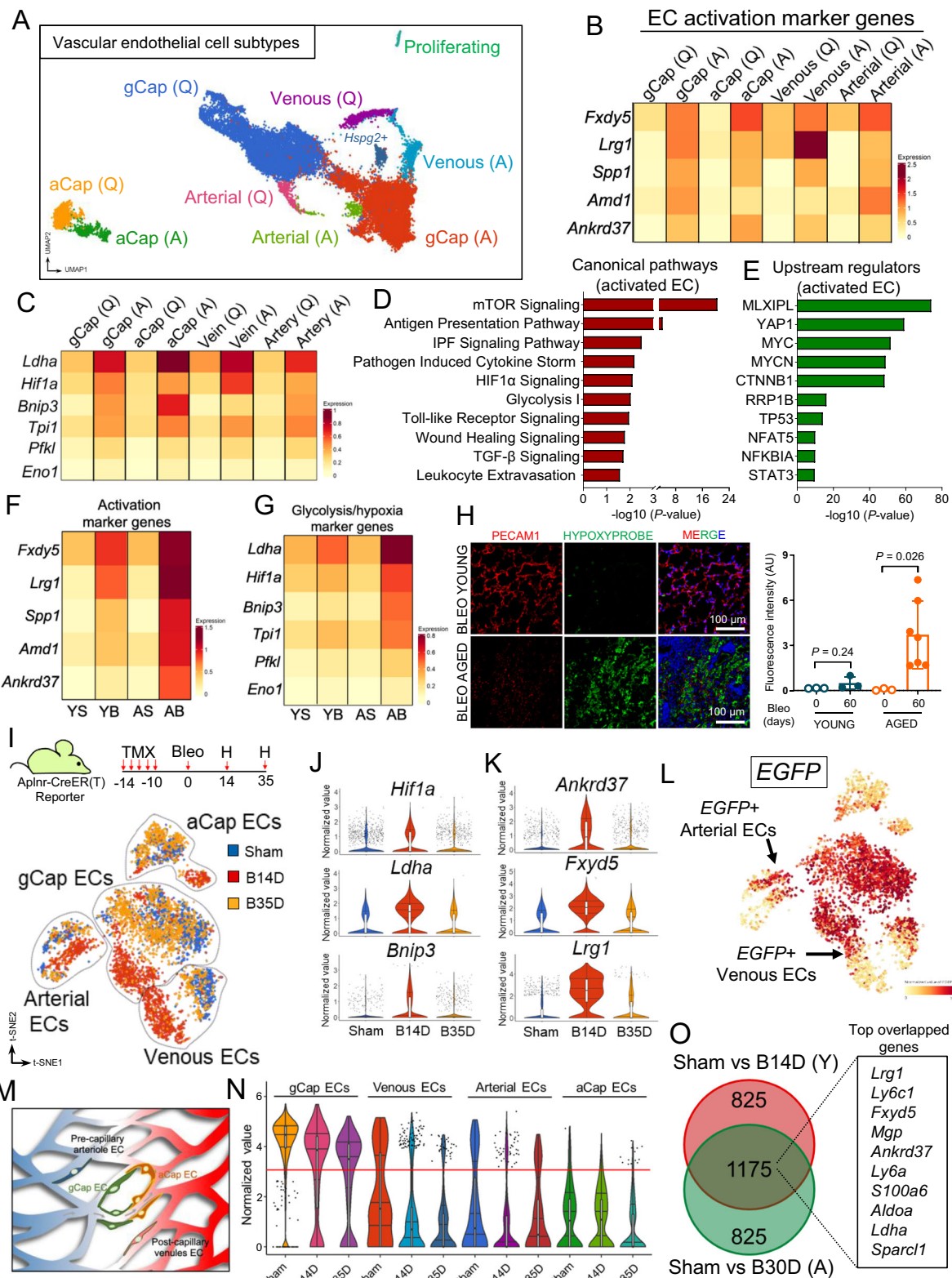

(day 35 days post bleomycin). For this experiment, we used a conditional *Aplnr-Cre-ER(T):Rosa26-mtdTomato/mEGFP* reporter mouse (Aplnr-Cre-ER(T)-mTmG), in which APLNR-expressing cells are permanently labeled with EGFP upon tamoxifen administration (Fig. 1I). *Aplnr* gene was previously reported to be expressed in lung ECs derived from the microcirculation[17], including those from post-capillary venules[29]. This reporter mouse enables us to simultaneously trace the fate of capillary lung ECs while assessing temporal dynamics

of other lung ECs during fibrotic peak and resolution. As shown in Fig. 1I, several EC clusters emerged following lung injury which largely segregate based on EC identity (capillary-, arterial-, and venous-derived ECs) and magnitude of injury-induced activation. ECs derived from lungs harvested at the fibrotic peak (day 14) were transcriptionally distinct from their quiescent counterparts (red clusters vs blue clusters). On the contrary, ECs derived from lungs isolated during the early resolution phase of fibrosis (day 35) were transcriptionally

**Fig. 1 | Activation of mouse lung ECs following bleomycin challenge. A** UMAP embedded visualization of different EC subpopulations (*n* = 24,168 cells). **B** Heatmap showing differentially expressed activation marker genes in each EC subpopulation. **C** Heatmap showing differentially expressed hypoxia/glycolysis marker genes in each EC subpopulation. **D, E** Ingenuity pathway analysis shows canonical pathways and upstream regulators enriched in all activated ECs. *P* values were generated using Right-tailed Fisher's Exact Test (log2 FC ≤ −0.5 or ≥0.5, *P* value ≤ 0.05). **F** Heatmap showing differentially expressed activation marker genes in each experimental group. Young Sham (YS, *n* = 5699 cells); Young Bleo (YB, *n* = 2208 cells); Aged Sham (AS, *n* = 6810 cells); Aged Bleo (AB, *n* = 9451 cells). **G** Heatmap showing differentially expressed hypoxia/glycolysis marker genes in each experimental group. **H** Hypoxia detection by hypoxyprobe in young and aged mouse lungs at 60 days post bleomycin administration. Immunofluorescence staining using antibodies against Pimonidazole adducts and PECAM-1 shows reduced number of alveolar ECs in fibrotic aged lungs exhibiting elevated hypoxia compared to young one in which hypoxia levels were undetectable. Values are summarized as mean ± SEM and analyzed using a two-tailed Student's *t*-test. Young Bleo day 0 (*n* = 3), Young Bleo day 60 (*n* = 3), Aged Bleo day 0 (*n* = 3), Aged Bleo day

60 (*n* = 7). **I** Schematic showing the experimental strategy and t-SNE displaying different EC subpopulations in sham (Blue, *n* = 3), injured lungs after 14 days (B14D, Red, *n* = 2), and injured lungs at 35 days (B35D, Orange, *n* = 2;) post bleomycin-induced lung injury. **J, K** Violin plots showing increased expression of activation and hypoxia/glycolysis marker genes in injured aged lungs at 14 days followed by a return to baseline during the early resolution phase (day 35 post bleomycin challenge), sham (*n* = 1994 cells), B14D (*n* = 2164 cells), and B35D (*n* = 1807 cells). **L** t-SNE plots showing the expression of *EGFP* across different EC clusters (*n* = 5965 cells). **M** Schematic showing arterial and vein EC subtypes expressing capillary markers. **N** Violin plot showing the expression of EGFP across different EC subpopulations (*n* = 5965 cells). **O** Venn diagram shows nearly 60% transcriptional overlapping between EC states at day 14 post bleomycin (young lungs (Y)) and day 30 post bleomycin (aged lungs (A)). The top two thousand differentially expressed genes (*P* < 0.05, determined by BioTuring Browser 3) were included in this analysis. Each box plot displays the median value as the center line, the upper and lower box boundaries at the first and third quartiles (25th and 75th percentiles), and the whiskers depict the minimum and maximum values. Source data are provided as a Source Data file.

similar to quiescent ECs (orange clusters vs blue clusters), supporting the concept that lung EC activation is reversible in young mice. Transcriptional analysis also confirmed the transient activation of lung ECs during the fibrotic peak and their return to quiescence during fibrosis resolution (Fig. 1J, K).

Intriguingly, although EGFP was primarily expressed by gCap ECs, as previously reported[17], we were also able to detect EGFP expression in other subpopulations of ECs derived from arterial and venous lineages (Fig. 1L). Notably, these EGFP-expressing arterial- and venous-derived ECs share numerous marker genes with gCap ECs (Supplementary Fig. 6), suggesting that these lung ECs subtypes were derived from the arterial and venous microcirculation (pre-capillary arteriole and post-capillary venules) (illustrated in Fig. 1M). Upon bleomycin injury, EGFP expression was largely retained in all activated capillary ECs during the fibrotic peak as well as during the early resolution phase when ECs have returned to quiescence (Fig. 1N), further highlighting the reversible nature of lung capillary EC activation following injury. Moreover, our results obtained using two independent scRNA-seq datasets demonstrated that activated ECs from young lungs at the peak of fibrosis (day 14 post bleomycin) were transcriptional similar to those from aged lung ECs during fibrosis resolution (day 30 post bleomycin) as indicated by the large number of dysregulated genes (Fig. 1O) and pathways (Supplementary Fig. 7) that were shared between these cellular states. Taken together, these data suggest that lung injury induces multiple metabolically active EC populations that may have roles in lung repair and fibrosis, and that aging perpetuates injury responses that are typically associated with the acute phase post lung injury.

**Compromised venous EC dynamics in fibrotic aged mouse lungs**
Venous endothelial cells have largely been studied in the context of tissue inflammation and immune cell recruitment both in vitro and in vivo[30]. The contribution of venous ECs to lung repair, and more broadly to organ fibrosis, however, is less known. Pulmonary venous ECs are heterogeneous, and their properties and functions vary based on their anatomic location (bronchial vs pulmonary circulation), and position in the vascular tree (veins vs venules)[31–33].

To study the impact of aging and bleomycin-induced lung injury on pulmonary venous EC responses, we sub-clustered venous ECs based on the expression of previously identified markers, including *Bst1, Slc6a2,* and *Amigo2* (Supplementary Fig. 3). Graph-based cluster analysis identified four transcriptional distinct venous EC subtypes (Fig. 2A): clusters 1 and 2 mainly included venous ECs from uninjured lungs (young and aged) and venous ECs from injured lungs that have returned to quiescence. Clusters 3 and 4 were occupied by activated venous ECs from injured lungs (young and aged) (Fig. 2B, C).

Intriguingly, the number of activated venous ECs in cluster 3 and 4 largely derived from injured aged lungs (Fig. 2C), suggesting that, unlike venous ECs from young lungs, venous ECs from aged lungs retained an activated state after injury. Gene enrichment analysis identified *Slc6a2* as a pan-venous gene marker that was expressed in all venous ECs (Fig. 2D), *Cyp4b1* and *Cpe* were identified as distinctive marker genes of quiescent venous ECs in clusters 1 and 2 respectively, and *Slc6a6* and *Ackr1* were identified as distinctive marker genes of activated venous ECs in clusters 3 and 4 respectively. *Slc6a6*+ activated venous ECs (cluster 3) shared many genes with *Cyp4b1*+ quiescent venous ECs (cluster 1), and among them were numerous marker genes of capillary ECs, such as *Npr3* and *Sema3c*, suggesting that *Slc6a6*+ activated venous ECs derived from pre-existing *Cyp4b1*+ post-capillary venules. This transcriptional analysis of lung venous ECs also uncovered a unique gene expression signature in aged venous ECs (cluster 4), of which *Ackr1* was the most distinctive gene marker (Fig. 2D). *Ackr1* encodes for Atypical Chemokine Receptor-1 (ACKR1), a G protein-coupled receptor with a major role in chemokine sequestration, degradation, and transcytosis[34,35]. ACKR1 was also shown to regulate chemokine bioavailability and, consequently, leukocyte recruitment[36,37]. We identified numerous inflammatory-related genes that were highly expressed in *Ackr1*+ venous ECs, including *Selp* and *Tifa* (Fig. 2E). *Selp*, encodes for p-selectin, a membrane receptor that mediates the interaction between activated ECs and leukocytes[38], whereas *Tifa* encodes for TRAF Interacting Protein with Forkhead Associated Domain, an adapter protein that plays a key role in the activation of NF-kappa-B signaling pathway and inflammation[39].

We also found that *Ackr1*+ venous ECs shared numerous marker genes with *Fabp4*+ venous ECs (cluster 2), including *Il1r1, Ptgs2, and Serpine1* (Fig. 2E), suggesting that this newly emerged venous EC population derived from preexisting lung venous ECs with specialized inflammatory functions. This hypothesis was supported by IPA and upstream regulator analysis on *Ackr1*+ venous ECs showing that genes implicated in multiple inflammatory pathways were enriched in this cluster (Fig. 2F, G). In addition, IPA pathway analysis identified several signaling pathways and regulators that are enriched in *Ackr1*+ venous ECs, including those implicated in vessel remodeling and angiogenesis, such as Rho family GTPase and VEGF pathways (Fig. 2F). These findings indicate that lung venous ECs exhibiting inflammatory and angiogenic features persisted in fibrotic aged mouse lungs, implicating lung venous remodeling as a previously unappreciated phenomenon with potential repercussions on the development of fibrosis.

To specifically investigate venous EC dynamics during the fibrotic peak and the resolution phase of bleomycin-induced lung fibrosis, we interrogated our longitudinal lineage tracing scRNA-seq dataset. As shown in Fig. 2H, unbiased cluster analysis identified multiple venous

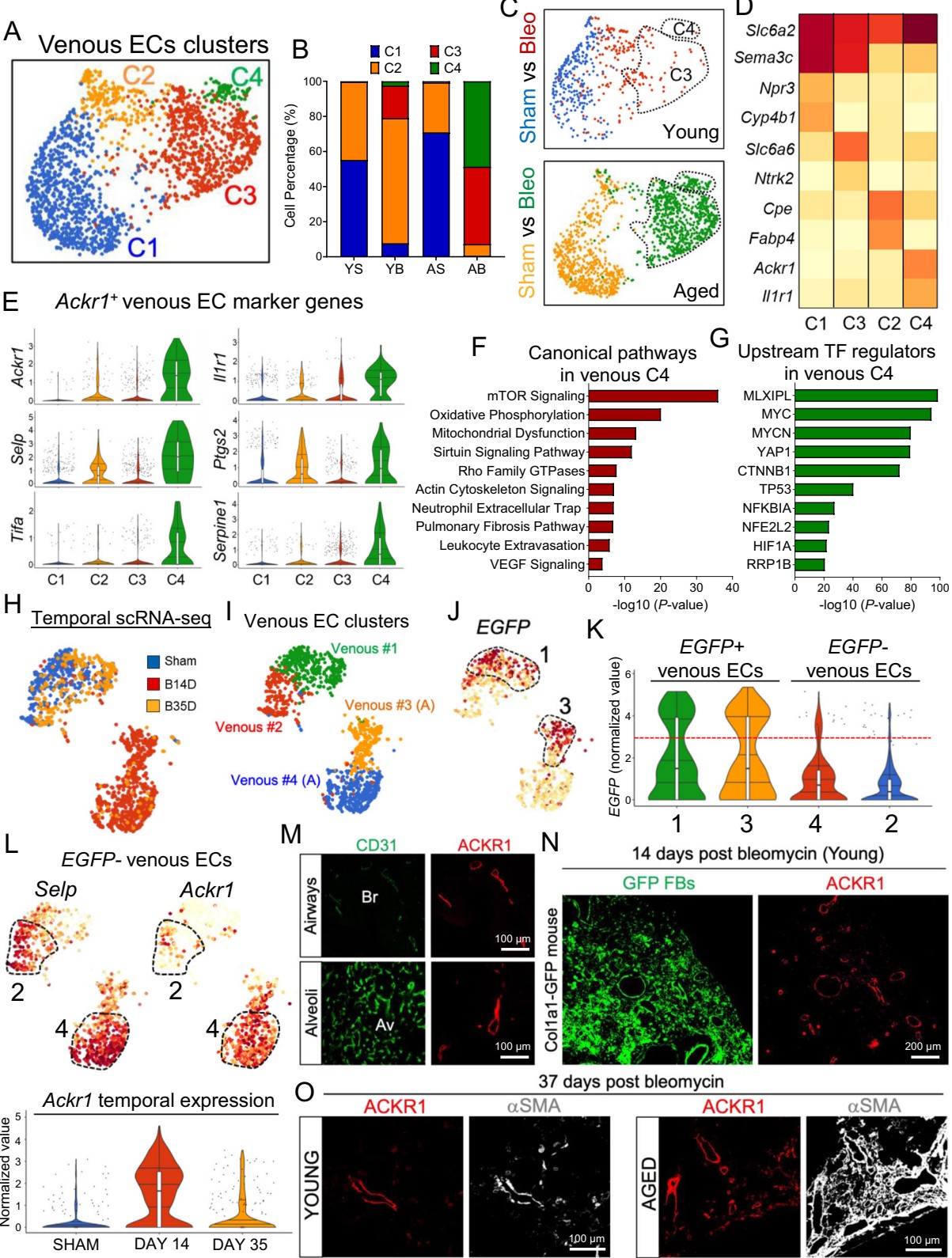

EC subtypes which separate based on EC identity and magnitude of cell activation. Venous ECs derived from lungs harvested during the fibrotic peak (day 14) (red cluster) were transcriptionally distinct from their quiescent counterparts (Sham) (blue cluster). On the contrary, venous ECs derived from lungs isolated during the early resolution phase of fibrosis (day 35) were transcriptionally similar to quiescent ECs (orange clusters vs blue clusters), further supporting the concept that lung EC activation is reversible in young mice.

Using these data, we also confirmed the existence of a sub-population of venous ECs (cluster #1) (Fig. 2I) that shared numerous marker genes of capillary EC, including *Gpihbp1* and *Npr3* (Supplementary Fig. 6) and was EGFP positive (Fig. 2J), supporting their post-

**Fig. 2 | Venous EC remodeling in young and aged mouse lungs following bleomycin challenge. A** UMAP plots showing venous EC clusters ($n = 1961$ cells). C; cluster. **B, C** Composition and UMAP plots displaying venous EC clusters in young (YS, Blue, $n = 302$ cells) and aged (AS, Orange, $n = 658$ cells) uninjured lungs, and young (YB, Red, $n = 199$ cells) and aged (AB, Green, $n = 802$ cells) injured lungs. Venous ECs in Clusters 3 and 4 exclusively emerged in bleomycin-injured lungs and their number increased in fibrotic aged lungs compared to young ones. **D** Heatmap showing average expression of venous EC marker genes across different venous EC clusters (**E**) Violin plots showing the expression of distinctive marker genes in activated venous ECs (clusters 4). C1 ($n = 873$ cells), C2 ($n = 231$ cells), C3 ($n = 778$ cells), C4 ($n = 79$ cells). **F, G** Ingenuity pathway analysis shows canonical pathways and upstream regulators enriched in activated venous ECs (clusters 4). $P$ values were generated using Right-tailed Fisher's Exact Test (log2 FC ≤ −0.5 or ≥ 0.5, $P$ value ≤ 0.05). **H, I** UMAP embedded visualizations of different venous EC clusters during different time point of bleomycin challenge, sham ($n = 383$ cells), bleomycin

14 days (B14D, $n = 703$ cells), bleomycin 35 days (B35D, $n = 253$ cells), Venous #1 ($n = 404$ cells), Venous #2 ($n = 229$ cells), Venous #3 ($n = 271$ cells), Venous #4 ($n = 435$ cells). A; Activated. **J, K** UMAP and violin plots showing the expression of *EGFP* across different venous EC clusters ($n = 1339$ cells). **L** UMAP and violin plots showing the expression of *Selp* and *Ackr1* genes across different venous EC clusters ($n = 1339$ cells). **M** Immunofluorescence staining showing ACKR1 (venous EC marker) and CD31 (Pan-endothelial marker) expression in lung airways and alveoli at baseline. Br; Bronchiole, Av; Alveoli. **N** Immunofluorescence staining showing ACKR1 expression in *Col1a1*-GFP mouse lung at 14 days after bleomycin injury. **O** Immunofluorescence staining using antibodies against ACKR1 and α-SMA in young and aged lungs following bleomycin injury. ACKR1 positive venous EC accumulates in lung areas exhibiting extensive remodeling. Each box plot displays the median value as the center line, the upper and lower box boundaries at the first and third quartiles (25th and 75th percentiles), and the whiskers depict the minimum and maximum values. Source data are provided as a Source Data file.

capillary venule origin. Moreover, cluster 3 (Fig. 2J, K), was largely occupied by activated *EGFP*-expressing venous ECs (day 14 post bleomycin), demonstrating that these activated venous ECs originated from post-capillary venules. Similarly, *EGFP* negative venous ECs expressing high levels of *Ackr1* and *Selp* genes appeared during the fibrotic peak and returned to baseline during the resolution phase (Fig. 2L). To shed further light into ACKR1+ venous EC remodeling associated with lung fibrosis, we immunostained normal and fibrotic mouse lungs with antibodies against ACKR1 and the pan-EC marker CD31. As shown in Fig. 2M, in normal mouse lungs, CD31 was expressed in alveolar capillary ECs as well as in ECs from larger vascular beds. On the contrary, ACKR1 expression was exclusively detected in lung ECs from large and small veins. ACKR1+ venous ECs were mainly found underneath the epithelial layer of both large bronchi and small bronchioles. Though sporadic, ACKR1+ venules were also identified in the alveolar space where they are intimately connected with CD31+ capillaries of the pulmonary circulation (Fig. 2M). Immunostaining analysis of bleomycin-treated lungs obtained from *Col1a1*-GFP mice showed that ACKR1+ venules were closely associated with distal alveolar regions exhibiting GFP+ fibroblast aggregation during the fibrotic peak (Fig. 2N). Furthermore, we found that ACKR1+ venous ECs were abundant in bleomycin-treated aged lungs compared to young ones and were closely associated with αSMA+ mesenchymal cells (Fig. 2O).

## ACKR1+ venous ECs expand in human fibrotic lungs

To translate our observations to human disease, we carried out immunostaining analysis on human lungs isolated from patients with IPF in parallel with lungs isolated from healthy donors. As shown in Fig. 3A, in normal human lungs ACKR1+ ECs were observed in small venules and veins around bronchi, bronchioles, and in the alveoli, and were focally surrounded by adventitial stromal cells expressing high levels of Collagen-I and αSMA. Intriguingly, Collagen-I + /αSMA+ stromal progenitor cells were recently identified in the peribronchial and alveolar regions of mouse and human lungs and were shown to give rise to pathogenic myofibroblasts in a mouse model of lung fibrosis[40], further suggesting that ACKR1 + EC/stromal cell crosstalk may be implicated in myofibroblasts appearance.

The vascular abnormalities described in IPF lungs have generated extensive debates and controversy over the last two decades[22,41,42]. Early studies had reported both capillary loss as well as increased vessels density in IPF lungs[9,22,43], suggesting that these divergent vascular abnormalities may co-exist in the fibrotic lung, and may reflect a diverse endothelial maladaptation to injury in different lung EC subtypes across different locations. To shed further light into the pathological vascular remodeling and spatial endothelial heterogeneity associated with human lung fibrosis, we first immunostained IPF lungs with antibodies against the pan-endothelial cell marker CD31. As shown in Fig. 3B, CD31 staining highlighted ECs across different lung

locations, including small alveolar capillaries and large vessels around bronchi. Notably, the alveolar parenchyma in these diseased lungs was largely replaced by vascularized fibrotic tissue, which enveloped terminal bronchi and extended diffusely into the adjacent lung parenchyma resulting in distortion and replacement of alveolar structures (Fig. 3B). Compared to intact parenchyma, areas of fibrosis contained a reduced number of vessels which were larger than the capillaries present in the alveolar septa. Trichrome staining and immunofluorescence analysis further showed that most newly formed vessels within this fibrotic lung tissue were positive for the venous cell marker ACKR1 (Fig. 3C, D). Furthermore, alveolar capillaries (CD31 + ACKR1-) were largely absent in this fibrotic tissue which was mainly populated by ACKR1+ ECs and αSMA+ stromal cells (Fig. 3E), demonstrating that, similarly to fibrotic aged mouse lungs, human IPF lungs were characterized by aberrant venous EC turnover. Intriguingly, ACKR1 marks venous ECs of both bronchial and alveolar circulation (Fig. 3A), suggesting that ACKR1+ venous ECs in IPF lungs may originate from either vascular bed.

Our immunostaining analysis of IPF lungs also revealed the presence of several vascular abnormalities involving venous ECs. For example, we detected ACKR1+ venous channels in the fibrotic intima of numerous arteries (Supplementary Fig. 8). These fibrotic vessels also showed medial thickening and αSMA+ cells associated with ACKR1+ venous ECs within the thickened intima, demonstrating that aberrant venous remodeling in IPF affects multiple lung districts.

## ACKR1+ venous ECs and pathogenic fibroblasts coexist in fibrotic areas of IPF lung

To link the accumulation of venous ECs to the degree of fibrosis, we evaluated the number of venous ECs and fibroblasts in various regions of human IPF lungs or healthy lungs using Fluorescence-Activated Cell Sorting (FACS) analysis (Fig. 4). Based on gross tissue assessment, we selected three lung regions with various degrees of fibrosis consolidation (high, medium, and low), and carried out a FACS analysis using antibodies against the pan-endothelial cell marker CD31, the venous markers ACKR1 and p-Selectin, and the mesenchymal cell markers Thy-1 and CTHRC1. Of note, Thy-1 was previously identified as a pan-fibroblast marker in the lung, and CTHRC1 was found to be highly expressed in scar-forming fibroblasts in fibrotic mouse and human lungs[40,44]. First, we found that the number of CD31+ cells was reduced in fibrotic areas (4.8%) relative to healthy-looking ones (10%) (Fig. 4A), supporting our previous findings that lung ECs are reduced in IPF lungs[8,9], and that this abnormality inversely correlated with the magnitude of fibrosis. We also found that although the overall number of CD31+ ECs was reduced in fibrotic areas, the number of venous ECs (ACKR1 + P-selectin + ) was relatively higher in these areas compared to less fibrotic ones (Fig. 4B), and this anomaly was accompanied by an increased number of Thy-1- and CTHRC1-expressing fibroblasts (Fig. 4B, C).

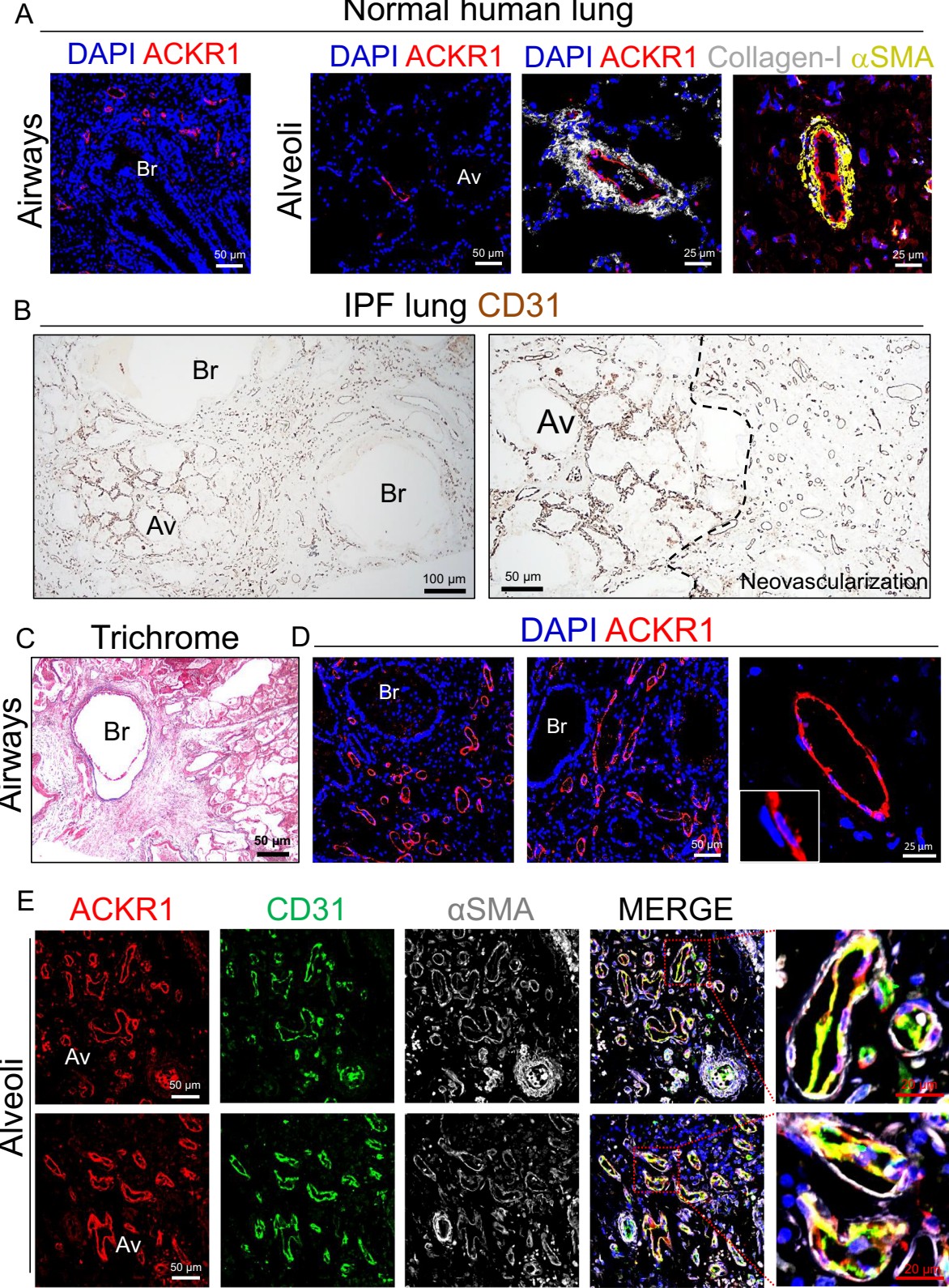

**Fig. 3 | ACKR1+ venous ECs expand in human fibrotic lungs.**
**A** Immunofluorescence staining of normal human lungs using antibodies against ACKR1, Collagen-I, and αSMA. ACKR1 is mainly expressed in peribronchial and alveolar venous ECs. Perivascular cells surrounding ACKR1 positive venous ECs strongly expressed Collagen-I and αSMA. Br Bronchiole, Av Alveoli.
**B** Immunohistochemistry of human IPF lung sections showing normal-looking alveoli surrounded by highly vascularized (CD31 positive) fibrotic areas. **C** Masson trichrome staining of human IPF lung sections showing peribronchial fibrosis.
**D** Immunofluorescence staining of human IPF lung sections showing ACKR1+ veins that surround a fibrotic small bronchus extend toward the alveolar parenchyma.
**E** ACKR1 positive veins with αSMA+ mural cells are found in the fibrotic alveolar parenchyma.

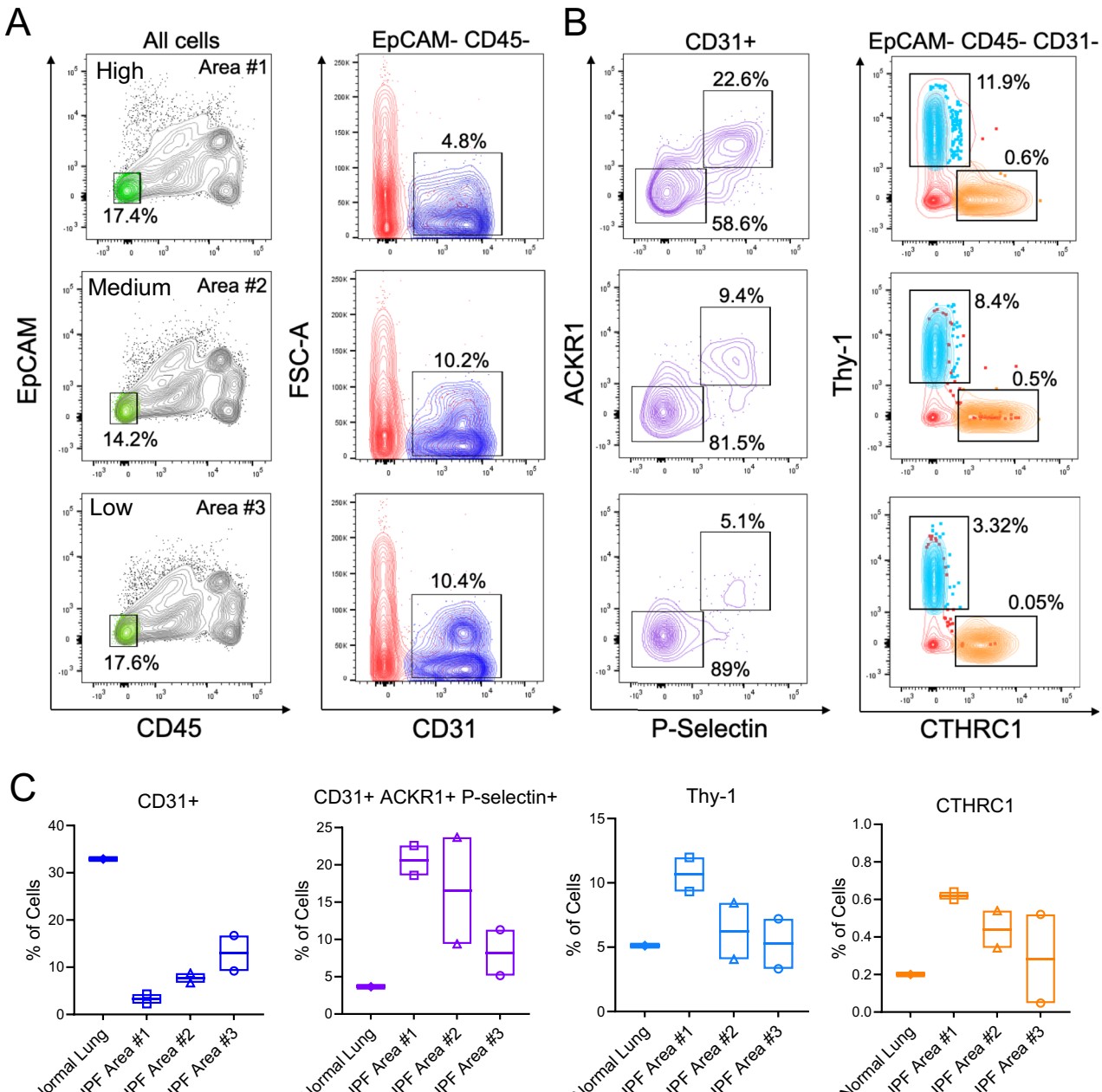

**Fig. 4 | ACKR1+ venous ECs expand in lPF lung regions containing elevated number of pathogenic fibroblasts.** FACS analysis was performed on human normal and IPF lung tissues. **A** Flow cytometry plot analyses are shown across different regions of fibrotic lungs using antibodies for the epithelial cell marker EpCAM, the immune cell marker CD45, the endothelial cell marker CD31, and (**B**) the venous EC markers ACKR1 and P-Selectin. **C** Quantitation of flow cytometry data demonstrates that ACKR1 + P-Selectin+ venous ECs were enriched in fibrotic areas compared to normal tissues containing a high number of pathogenic fibroblasts (Thy1 + CTHRC1 + ). One section from one normal lung and two sections from three different areas of one IPF lung were analyzed. Box borders display the minimum and maximum values, with a central line at the mean. Source data are provided as a Source Data file.

## Aberrant capillary ECs persist in aged lungs following injury

We previously showed that capillary EC repair after lung injury is impaired in aged lungs[8,9]. To investigate cellular and molecular mechanisms implicated in aging-associated aberrant lung capillary EC repair post bleomycin challenge, we re-clustered lung capillary ECs (gCap ECs and aCap ECs) using previously identified marker genes[17] (Fig. 5A). Cluster analysis revealed 7 distinct capillary EC clusters across all groups, 4 clusters expressing gCap EC marker genes and 3 expressing aCap EC marker genes. These clusters can be grouped into quiescent (Q) gCap and aCap ECs, which are predominant in the sham groups (young and aged), and activated (A) gCap (1, 2, and 3) and aCap

(1 and 2) ECs, which exclusively emerged in bleomycin-treated groups (young and aged) (Fig. 5B, C). Intriguingly, though activated capillary ECs emerged in both young and aged lungs post injury, their cellular composition and transcriptional signatures varied considerably with aging. In fact, gCap ECs in cluster 1 largely derived from injured young lungs, whereas those in clusters 2 and 3 are primarily constituted by gCap ECs derived from injured aged lungs (Fig. 5B). Although gCap ECs in cluster 1 share some marker genes of EC activation with gCap ECs in clusters 2 and 3, such as *Ntrk2* and *Lrg1* (Fig. 5D), they were transcriptionally similar to quiescent gCap ECs, as evidenced by the elevated expression of canonical gCap EC marker genes, such as *Plvap*

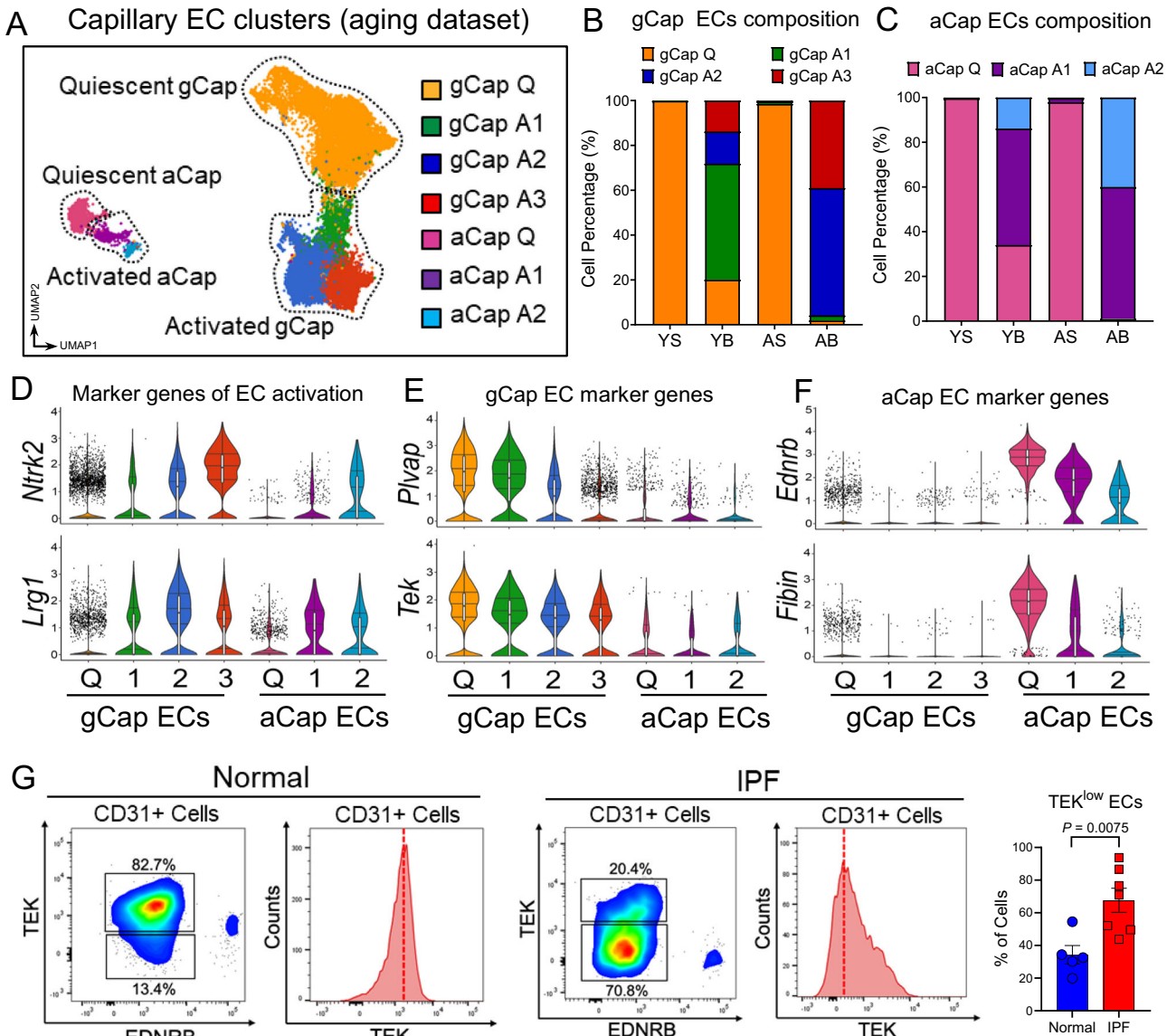

**Fig. 5 | Lung injury promotes capillary EC transition to activated/immature states that persist in aged and IPF lungs. A** UMAP plot of capillary ECs (n = 20960). Each color indicates a distinct EC state at baseline and in response to injury. Quiescent (Q), activated (A). **B**, **C** Composition of gCap and aCap EC clusters across different groups. Young Sham (YS, n = 5206 cells); Young Bleo (YB, 1867 cells); Aged Sham (AS, n = 5972 cells); Aged Bleo (AB, n = 8015 cells). **D** Violin plots showing the de novo expression of activated EC marker genes across different gCap (Q, (n = 10256 cells), 1 (n = 4354 cells), 2 (n = 3070 cells), 3 (n = 1121 cells)) and aCap EC (Q (n = 1283 cells), 1 (n = 545 cells), 2 (n = 331 cells)) clusters. **E** Violin plots showing the expression of quiescent gCap EC marker genes across different clusters. Activated gCap EC in cluster 2 and 3 exhibit the strongest reduction of canonical gCap EC marker genes. **F** Violin plots showing the expression of

quiescent aCap EC marker genes across different clusters. Activated aCap EC (cluster 1 and 2) exhibit the strongest reduction of canonical aCap EC marker genes. Each box plot displays the median value as the center line, the upper and lower box boundaries at the first and third quartiles (25th and 75th percentiles), and the whiskers depict the minimum and maximum values. **G** FACS analysis of normal (n = 5) and IPF (n = 7) lungs. Antibodies against TEK and EDNRB were used to discriminate gCap EC from aCap EC. Out of all CD31 positive cells gCap ECs were defined as (TEK^High EDNRB^Low) whereas aCap ECs strongly expressed EDNRB and were largely negative for TEK. IPF lungs exhibited an increased percentage of ECs expressing low levels of the gCap marker TEK compared to healthy lungs. Values are summarized as mean ± SEM and analyzed using a two-tailed Student's t-test. Source data are provided as a Source Data file.

and *Tek* (Fig. 5E). Though less prominent, we also found that the distribution of aCap ECs varied between young and aged lungs as evidenced by the increased number of activated aCap ECs expressing low level of the marker genes *Ednrb* and *Fibin* in injured aged lungs (Fig. 5C, F). Collectively, these findings demonstrated that lung injury in aged mice results in the accumulation of activated/immature capillary ECs that, unlike young mice, fail to return to quiescence.

To extend these findings to human disease and investigate the capillary EC turnover in the context of progressive lung fibrogenesis, we carried out a FACS analysis on diseased lungs from patients with IPF (Fig. 5G). We developed a FACS strategy using antibodies against

markers that were enriched in capillary ECs relative to ECs from other vascular beds, including TEK (gCap ECs), EDNRB (aCap ECs), and CD31 (pan-ECs marker). Using this strategy we discovered that within each CD31 + EC population, the number of ECs expressing the gCap EC marker gene *TEK* was significantly lower in IPF relative to healthy lungs, suggesting that, similarly to fibrotic aged mouse lungs, human fibrotic lungs exhibited an increased number of immature gCap ECs.

**TrkB expression marks capillary EC activation in injured lungs**
Previous studies have suggested that lung gCap ECs can give rise to aCap ECs under certain injury conditions[17]. While our scRNA-seq

analysis showed that activated gCap ECs and activated aCap ECs exhibited distinct gene expression signatures, they also revealed that they share multiple marker genes of EC activation, including *Ntrk2 and Lgr1* (Fig. 5D). Notably, among these, *Ntrk2* was the only marker gene that was mainly expressed by activated gCap ECs (Supplementary Fig. 9). *Ntrk2* encodes for Tropomyosin receptor kinase B (TrkB), a membrane receptor that has been previously implicated in neurite outgrowth and angiogenesis[45,46].

To investigate the fate and temporal dynamics of gCap ECs post bleomycin challenge, we interrogated our longitudinal lineage tracing scRNA-seq dataset and specifically focused on capillary EC subtypes. Unbiased cluster analysis identified 6 distinct capillary EC clusters, which segregate based on endothelial cell identity (gCap ECs vs aCap ECs) and time post injury (day 14 vs day 35) (Fig. 6A). Cluster and gene expression analysis showed that *Ntrk2* gene was highly expressed in activated capillary ECs at day 14 post injury and its expression returned to baseline during the early resolution phase (day 35 post injury) (Fig. 6B, C). Permanently labeled *EGFP* + ECs were found in quiescent gCap ECs as well in activated gCap ECs at day 14 and day 35 post bleomycin challenge (Fig. 6D, E), thus confirming the reversible nature of capillary EC activation post injury. Remarkably, the number of aCap ECs expressing *EGFP* was nearly undetectable (Fig. 6E), demonstrating that, under these experimental conditions, gCap ECs do not acquire transcriptional features of aCap ECs. To validate these findings at protein level, we immunostained the lungs of our gCap EC reporter mice (sham and 28 days post injury) using an antibody against the aCap EC marker CAR4. Data displayed in Fig. 6F shows minimal or no overlapping between EGFP and CAR4 positive cells in absence of injury, and this expression pattern was maintained after bleomycin injury.

Our single cell lineage tracing model gives us the opportunity to examine whether capillary ECs acquire mesenchymal features during the progression of lung fibrosis. Previous studies have reported that ECs undergo mesenchymal transition during lung fibrosis to become collagen producing myofibroblasts[47,48]. To investigate capillary ECs transition toward a mesenchymal fate, we sub-clustered endothelial (*Cdh5* +) and mesenchymal cells (*Col1a1* +) (Supplementary Fig. 10A, B) and interrogated this subcluster for *EGFP* expression. As shown in Supplementary Fig. 10C, D *EGFP* was mainly expressed in gCap ECs, and, to less degree, in ECs of venous and arterial origin, however, no *EGFP* expression was detected in cells of the mesenchymal clusters. Although our lineage tracing study demonstrated that injured capillary ECs transiently overexpressed mesenchymal-associated genes, such as *Col1a1*, *Acta2*, and *Fn1* during the fibrotic peak (Supplementary Fig. 10E), these cells largely maintained their endothelial identity. Collectively, these data demonstrated that activated lung capillary ECs, which transitionally arise after injury, and accumulate in fibrotic aged lungs, are not the result of an aberrant differentiation program but rather a manifestation of their incapacity to return to quiescence.

To validate the emergence of transitional TrkB+ gCap ECs post lung injury we immunostained the lungs of our reporter mice using an antibody against TrkB. As shown in Fig. 6H, TrkB was expressed in EGFP+ gCap ECs from injured lungs but was undetectable in EGFP+ gCap ECs from uninjured lungs, demonstrating the TrkB is expressed de novo in capillary ECs in response to lung injury. Furthermore, immunostaining analysis of young lungs from bleomycin-treated Col1a1-GFP mice showed that TrkB+ capillary ECs emerge during the fibrotic peak (day 14 post bleomycin) and they closely associated with injured lung areas exhibiting increased fibroblast aggregation (Fig. 7A). While TrkB+ capillary ECs were still detected during the early resolution phase of lung fibrosis (day 37 post bleomycin), these cells were limited to lung areas exhibiting residual fibroblast aggregation.

Next, to assess the impact of aging on the appearance and tissue localization of TrkB+ capillary EC following injury, we immunostained the lungs of young and aged mice during the early resolution phase of

bleomycin-induced lung injury (37 days post-bleomycin) using antibodies against TrkB and Collagen-I. As shown in Fig. 7B, TrkB+ cells were detected in both injured young and aged lungs during this reparative phase, though, these intermediate cells were more abundant in aged lungs in areas exhibiting robust collagen-I deposition, suggesting a positive correlation between the number of TrkB+ cells and the magnitude of lung fibrosis. To translate these finding to human disease, we immunostained lung tissues from patients with IPF or healthy donors using antibodies against TrkB and CD93 (capillary marker). As shown in Fig. 7C, healthy lungs exhibited low expression of TrkB in CD93+ capillary ECs, whereas IPF lungs exhibited increased number of capillary EC expressing TrkB which were mainly located around fibroblastic foci (FF). Altogether, these in vivo data substantiated our transcriptional analysis and highlighted TrkB+ gCap ECs as a population of capillary ECs that transiently emerged in injured young lungs and persisted in aged lungs with sustained fibrosis.

## YAP/TAZ and TrkB cooperate to regulate lung capillary EC morphogenesis

We previously identified YAP as a putative upstream regulator implicated in the transcriptional program leading to EC activation following bleomycin injury (Fig. 1E). Aside for its known function as mechano-regulator, YAP (encoded by *YAP1* gene) and its paralog TAZ (encoded by the *WWTR1* gene) have been implicated in metabolic reprogramming during cell growth and differentiation[49–51]. YAP/TAZ-mediated transcription has been shown to regulate various metabolic pathways, including glycolysis, mitochondrial respiration, and autophagy[52–55], which are required for cellular adaption to adverse environmental conditions, such as injury or oxygen deprivation. To study the contribution of YAP/TAZ activation or inhibition to capillary EC behavior, we treated human lung microvascular ECs (HLMVECs) with TRULI, a recently discovered small-molecule inhibitor of LATS kinases that promotes YAP/TAZ nuclear activity[56]. HLMVECs treated with this inhibitor exhibited increased expression of the activation marker gene *NTRK2*, and reduced expression of the canonical capillary markers *TEK*, *KIT*, and *PLVAP* (Fig. 8A). Inversely, YAP and TAZ knockdown in these cells inhibited *NTRK2* expression and upregulated *TEK*, *KIT* and *PLVAP* gene expression (Fig. 8B). These findings using human cells align with our observations in mice and support a role for YAP/TAZ pathway in capillary EC transition to an activated poorly differentiated state following lung injury. As activated TrkB+ capillary ECs largely accumulated in the lungs of aged mice with delayed fibrosis resolution, these findings suggest that aberrant YAP/TAZ signaling in aged gCap ECs may contribute to their sustained activation/de-differentiation, resulting in disrupted capillary homeostasis and persistent fibrogenesis.

The appearance of TrkB+ capillary ECs following lung injury and the previously described role of TrkB signaling pathway in vascular remodeling and angiogenesis[46,57], prompted us to further investigate the biological relevance of TrkB pathways in lung capillary morphogenesis. Previous studies showed that brain-derived neurotrophic factor (BDNF), the natural ligand of the TrkB receptor, plays role in alveolar repair after influenza[58], and identified lung epithelium as the main source of this secreted neurotrophic factor[58]. Motivated by these relevant findings, we sought to interrogate our scRNA-seq of young and aged mouse lungs for the expression of *Bdnf*. We found that *Bdnf* gene was strongly expressed in alveolar type-I-epithelial cells (ATI), and weakly or not expressed in other lung cell types, including alveolar type-II-epithelial cells (ATII) (Fig. 8C–E). To specifically assess the expression of alveolar epithelial *Bdnf* in the lungs of young and aged mice post bleomycin challenge, we re-clustered alveolar epithelial cells using previously identified ATI and ATII marker genes lineage markers. As shown in Fig. 8D, unbiased cluster analysis revealed three distinct population of alveolar epithelial cells: ATII, which was the most abundant alveolar epithelial cell cluster, ATI, and an intermediated

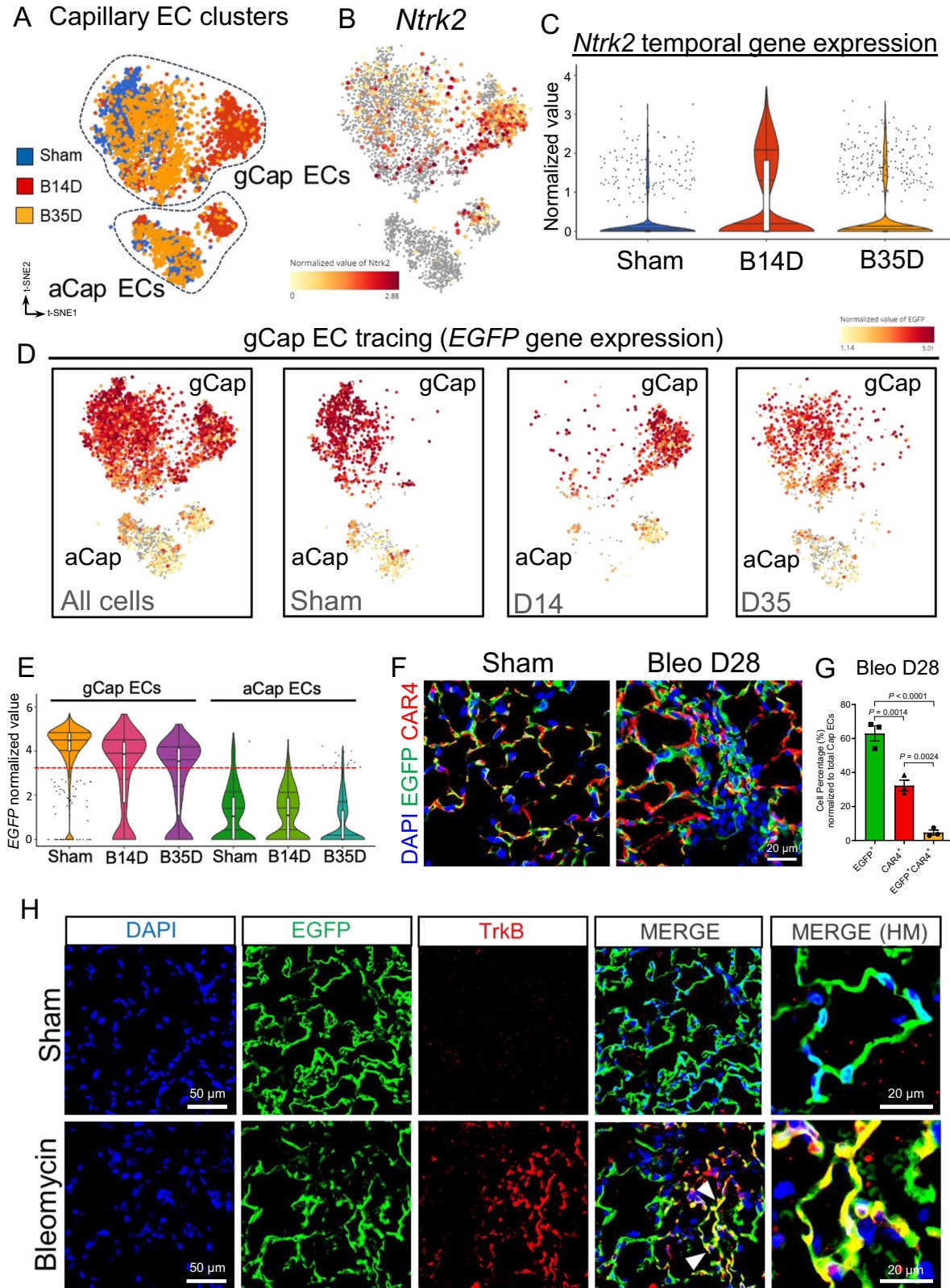

cluster of alveolar epithelial cells (here defined as intAE) that co-expressed both ATII and ATI markers. Notably, this intermediate cell population appeared almost exclusively in bleomycin treated lungs (young and aged) and was accompanied by the reduction of ATII progenitor cells (Supplementary Fig. 11A), suggesting that this intermediate cell population emerged from ATII cells as they differentiate into ATI cells. While this intermediate cell cluster shared numerous

markers with ATI cells, such as *Hopx*, *Vegfa*, and *Rtkn2*, it also expressed a distinct set of genes, including *Clu*, *Krt8*, and *Cdkn1a* that were weakly expressed by differentiated ATI or ATII cells (Supplementary Fig. 11B). These findings are consistent with previous observations of the emergence of intermediate Krt8+ ATII-derived cells in the lungs of mice treated with bleomycin[59]. To gain further insights into the impact of aging on the intermediate cell population we re-

**Fig. 6 | TrkB is a marker of activated capillary ECs in response to bleomycin challenge. A** t-SNE displaying different capillary EC subpopulations in sham (Blue, $n = 1346$ cells), injured lungs after 14 days (B14D, Red, $n = 1067$ cells), and injured lungs after 35 days (B35D, Orange, $n = 1399$ cells) post bleomycin-induced lung injury. **B**, **C** t-SNE and violin plots showing that the expression of *Ntrk2* gene is enriched in capillary ECs at 14 days after injury and returned to baseline at day 35 post bleomycin challenge. Sham ($n = 1346$ cells), B14D ($n = 1067$ cells), B35D ($n = 1399$ cells). **D**, **E** t-SNE and violin plots showing the expression of *EGFP* gene following injury. gCap ECs (sham ($n = 964$ cells), B14D ($n = 800$ cells), B35D ($n = 1002$ cells) aCap ECs (sham ($n = 382$ cells), B14D ($n = 267$ cells), B35D ($n = 397$ cells)). **F** Immunofluorescence staining using antibodies against GFP and CAR4 in young lungs of uninjured sham and 28 days after bleomycin injury. **G** Quantification of immunofluorescence staining at 28 days after bleomycin injury showing the

retention of EGFP expression by gCap ECs at different time points and the absence of EGFP expression in aCap ECs ($n = 3$). Values are summarized as mean ± SEM, *P* values were generated using one-way ANOVA with Tukey's post hoc test for comparison. **H** Immunofluorescence images showing the expression of TrkB in gCap ECs after bleomycin challenge. gCap ECs were lineage labeled in Aplnr-CreER(T)-mTmG mice 15 days prior to bleomycin administration (Day 0). Sham and bleomycin-injured lungs were harvested 28 days post bleomycin delivery followed by immunofluorescence analysis. An antibody against TrkB was used to detect injured gCap ECs (Red). gCap ECs co-expressing EGFP and TrkB (yellow) only emerged in injured lungs. Each box plot displays the median value as the center line, the upper and lower box boundaries at the first and third quartiles (25th and 75th percentiles), and the whiskers depict the minimum and maximum values. Source data are provided as a Source Data file.

clustered *Krt8+Cdkn1a+* intAE cells and obtained two distinct sub-clusters (Intermediate Cluster 1 and Intermediate Cluster 2) (Fig. 8F), with each sub-cluster expressing a unique set of genes, including *Spock2* and *Sdpr* (cluster 1) and *Scd1* and *Ccnd1* (cluster 2) (Fig. 8G). These findings suggested that following lung injury ATII cells pass through multiple intermediate transcriptional states before transitioning into fully functional ATI cells. Notably, *Bdnf* gene was highly expressed by epithelial cells exhibiting a more mature ATI state (cluster 1) (Fig. 8H). Cell composition analysis revealed that the number of mature ATI cells expressing *Bdnf* gene was strongly reduced in aged lungs post bleomycin challenge (Fig. 8I), implicating dysfunctional ATI turnover and reduced BDNF availability with aging.

ATI cells are lost in IPF lungs, and dysfunctional ATII/ATI differentiation has been reported as a pathological feature implicated in sustained lung fibrosis in IPF lungs[59–61]. To extend our mouse observations and assess the expression of BDNF in the lungs of patients with pulmonary fibrosis, we interrogated a publicly available scRNA-seq dataset from human IPF and healthy lungs[24]. This analysis showed that the expression of *BDNF* gene, specifically in ATI cells, was significantly reduced in the lungs of elderly patients with IPF compared to healthy lungs (Fig. 8J), suggesting that limited BDNF availability in IPF lungs may be implicated in dysfunctional endothelial cell remodeling/differentiation and abnormal repair. To study the functional role of BDNF/TrkB signaling pathway in lung capillary remodeling, we employed a capillary morphogenic assay in vitro (Fig. 8K). We rationalized that BDNF may promote capillary morphogenesis, and that such phenotypes may be elevated when capillaries are in an injury related "activated" state, a condition that we found is induced by TRULI treatment (Fig. 8A). In this assay human lung capillary ECs were embedded in a collagen-I gel in presence of endothelial cell media containing BDNF alone, TRULI alone, or both. Cells were cultured for 48 h to allow the formation of vessel-like structures. As shown in Fig. 8K, lung capillary ECs cultured in presence of TRULI alone exhibit a modest but significant increase in tube-like structures compared to control. BDNF alone showed no increase in the number of microvessels, however, this effect was overcome when TRULI was added in the culture media, suggesting a synergism between YAP and TrkB signaling pathways in regulating vessels remodeling. Taken together our observations suggest a disconnection between YAP and TrkB signaling in capillary ECs that occurs as a result of the loss of ATI cells, a phenotype that is exacerbated with aging.

## Discussion

The ability of lung to restore normal homeostasis after injury deteriorates with aging[1,2]. Dysfunctional or inadequate regeneration has been associated with disrepair and sustained lung scarring[62,63], suggesting that loss of the regenerative dynamics with aging contributes to perpetuating fibrogenic signals into the lung resulting in exuberant extracellular matrix deposition and deterioration of organ functions[1,13]. In this work, we investigated the cellular and transcriptional alterations accompanying persistent lung fibrosis in aged mice following

bleomycin injury and compared them to those observed in young mice in which fibrosis typically resolves. Our scRNA-seq analysis revealed cellular and transcriptional alterations in response to bleomycin injury that were aging associated. While we only profiled two young mouse lungs compared to four aged mouse lungs (a limitation of our study), we identified several EC populations that exhibited a shifted molecular state that we labelled as "activated" that persists in fibrotic aged and IPF lungs. We experimentally validated these activated EC sub-populations and showed that cells exhibiting these traits sit in close proximity to areas of collagen deposition, suggesting that signaling events within ECs may contribute to the fibrotic niche that persists with aging.

EC subpopulations were observed to acquire an activated state with injury, including venous and capillary lung ECs, and while lung EC activation resolved in young animals in concordance with fibrotic resolution, this activated state persisted in aged animals. We identified several marker genes, including *Lrg1*, *Spp1*, and *Ankrd37*, whose expression was associated with the EC activated states, as well as several glycolysis-associated genes. Our transcriptional analyses identified multiple pathways and upstream regulators associated with glycolysis including HIF1a, MYC and YAP/TAZ. Intriguingly, numerous HIF1α target genes were found to be upregulated in aged lung ECs compared to young ones, and among them were those implicated in glycolysis, hypoxia-mediated cell death, and organ fibrosis, such as *Ldha* and *Bnip3*[64–66]. These findings suggest that sustained HIF1α signaling in aged lung ECs may result in metabolic maladaptation to injury and impaired endothelial homeostasis. Indeed, we found that tissue hypoxia was significantly elevated in aged lungs with sustained fibrosis compared to young lungs, in which lung fibrosis gradually resolves. Of note, transient activation of hypoxia-mediated signaling pathways, such as HIF1α, has been shown to contribute to angiogenesis and vascular regeneration in multiple organs[67–70], but sustained tissue hypoxia due to aberrant vascular remodeling leads to disrupted tissue homeostasis, disrepair, and fibrosis[66,69,70].

Our single cell transcriptomic analysis revealed multiple sub-populations of ECs derived from venous EC lineages, including two that were associated with uninjured healthy mouse lungs. These observations suggest that the lung may possess distinct venous ECs that serve different functions in distinct districts of the pulmonary vasculature. The gene expression signatures of these venous ECs were profoundly altered following injury in both young and aged lungs as demonstrated by the acquisition of injury markers that were not expressed by quiescent venous ECs. Notably, injury activated genes, including *Spp1*, *Lrg1*, and *Ankrd37*, were also expressed by activated gCap ECs, suggesting that multiple endothelial lineages converge on common transcriptional states before returning to quiescence. Interestingly, we found that the transcriptional state of venous ECs that emerge with injury are different between young and aged lungs, with aged lungs exhibiting a distinct venous EC population characterized by the expression of the marker gene *Ackr1* and the upregulated expression of genes implicated in inflammation, suggesting that these cells

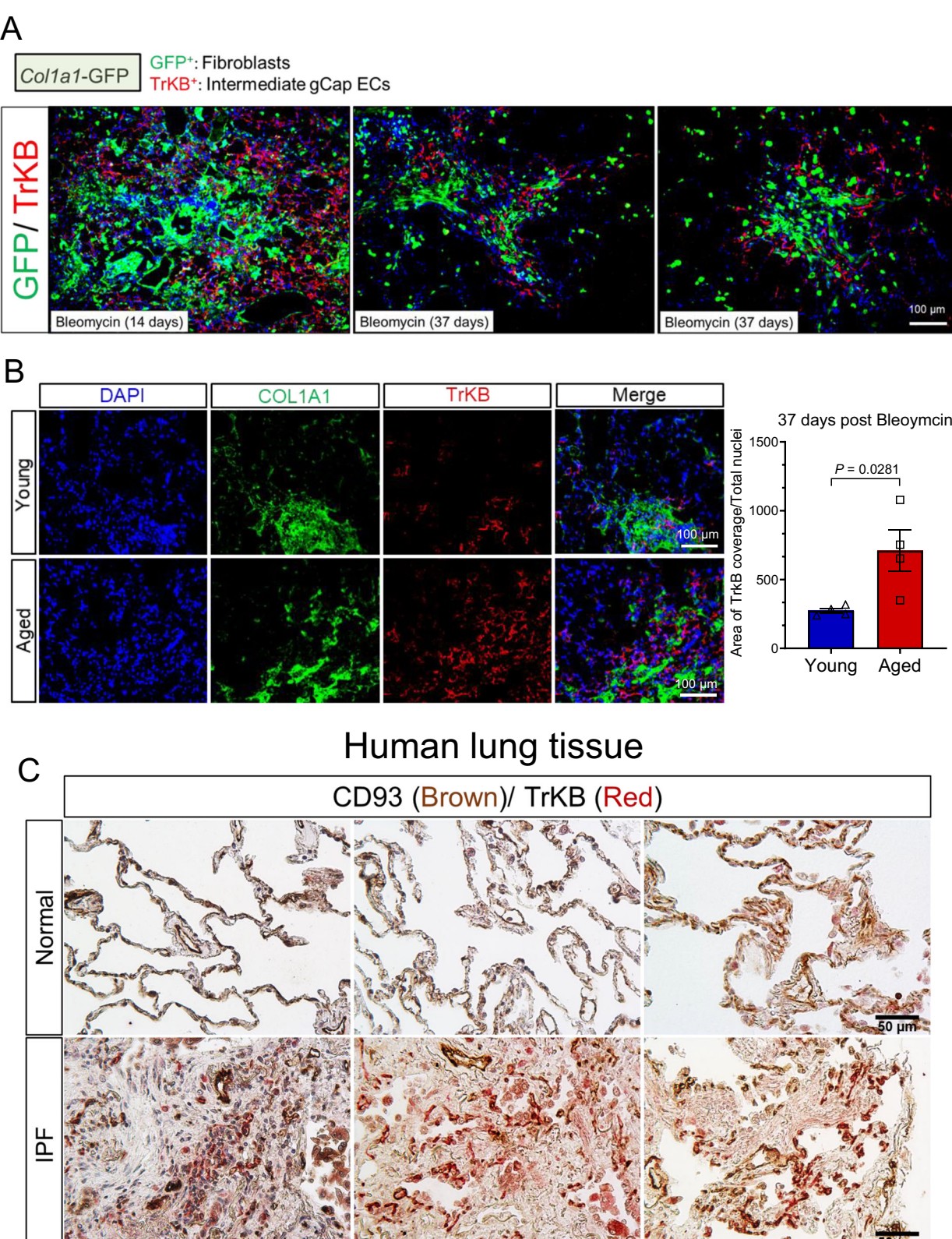

**Fig. 7 | TrkB+ gCap ECs accumulated in fibrotic aged mouse lungs and human IPF lungs. A** Immunofluorescence images showing the localization of TrkB+ gCap ECs in lung areas of fibroblast aggregation (GFP+ fibroblasts) at 14 and 37 days after bleomycin challenge, $n = 3$. **B** Immunofluorescence staining using antibodies against TrkB and collagen-I in young and aged lungs following bleomycin injury ($n = 4$). TrkB+ gCap ECs are localized in lung areas exhibiting collagen deposition. Values are summarized as mean ± SEM and analyzed using a two-tailed Student's *t*-test. Source data are provided as a Source Data file. **C** Images of normal ($n = 3$) and IPF ($n = 3$) human lungs stained by immunohistochemistry for TrkB and CD93.

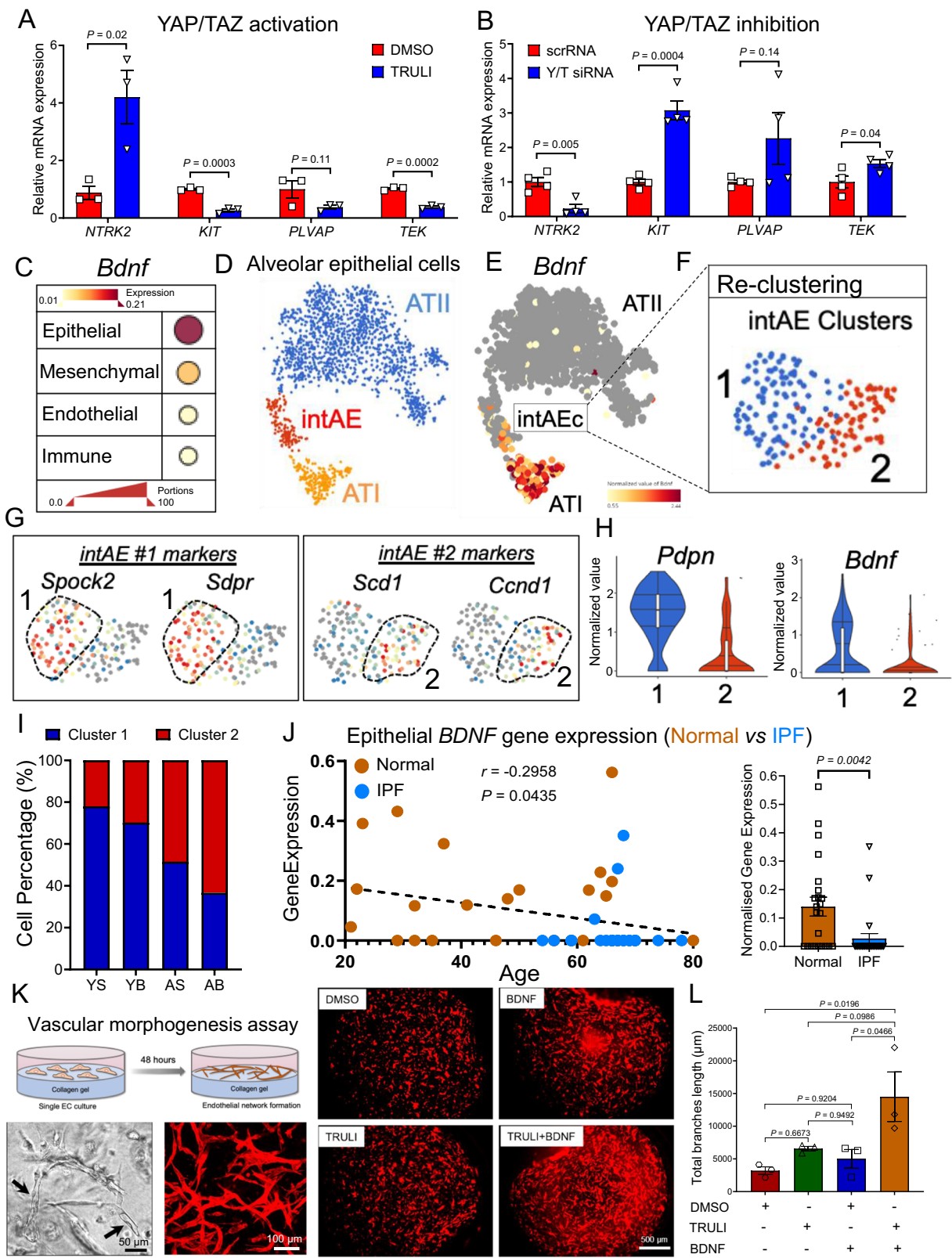

may orchestrate inflammatory responses that contribute to the lung fibrotic milieu. Consistent with the inflammatory activity of aged venous ECs, venous remodeling was mainly observed in dense fibrotic regions of aged mouse lungs, while this feature was less pronounced in injured young mice. Similarly, ACKR1+ veins were also abundant in fibrotic areas of IPF lungs where they were largely surrounded by αSMA+ cells, suggesting a pathogenic profibrotic function for this

venous EC population. This hypothesis was reinforced by our FACS analysis of IPF lungs demonstrating a positive correlation between the number of myofibroblasts and that of venous ECs in lung areas with different degrees of fibrosis. Recently, a population of venous ECs marked as *COL15a1* + , and whose location in healthy human lungs was restricted to bronchi and bronchioles[33], was found in distal alveolar regions of IPF lungs[71], suggesting that venous EC-mediated

**Fig. 8 | Activation of YAP/TrKB axis enhances lung capillary morphogenesis.**
**A**, **B** qPCR analyses of human lung microvascular ECs (HLMVECs) treated with the LATS1/2 inhibitor TRULI and siRNAs targeting YAP and TAZ for 48 or 72 h, respectively. YAP activation in these cells partially recapitulates the gene expression signature observed in activated gCap ECs. Values are summarized as mean ± SEM and analyzed using a two-tailed Student's *t*-test. Yap activation (*n* = 3 independent experiments), YAP/TAZ inhibition (*n* = 4 independent experiments). **C** Dot plot showing the expression of *Bdnf* in different populations of lung cells. Dot size indicates the proportion of expressing cells, colored by standardized expression levels. **D** t-SNE plot showing different alveolar epithelial cell clusters. ATII (Blue, *n* = 1470 cells), intermediate AE (intAE, Red, *n* = 176 cells), and ATI (Orange, *n* = 229 cells). **E** t-SNE plot showing the expression of *Bdnf* in intAE and ATI. **F**, **G** UMAP visualization of intermediated alveolar epithelial cell clusters and marker gene signatures. **H** Violin plots showing the expression of ATI cell marker gene, *Pdpn*, and *Bdnf* in intAE clusters, 1 (103 cells), 2 (73 cells). Each box plot displays the median value as the center line, the upper and lower box boundaries at the first and third quartiles (25th and 75th percentiles), and the whiskers depict the minimum and maximum values. **I** Cell composition in each intAE cluster. Most intermediate alveolar epithelial cells exhibiting ATII marker genes (less differentiated) were from bleomycin-treaded aged lungs. Young Sham (YS); Young Bleo (YB); Aged Sham (AS); Aged Bleo (AB). **J** Expression of BDNF ATI cells derived from human normal (*n* = 23) and IPF (*n* = 24) lung. Values are summarized as means ± SEM and analyzed using a two-tailed Student's *t*-test (**K**) Schematic showing the in vitro 3D endothelial morphogenesis assay. Staining of HLMVECs with phalloidin shows that BDNF and TRULI treatment synergistically promote tube formation. **L** Quantitative analysis of vascular morphogenesis was indicated as the total branches length. Image analysis in the whole photographed area were performed by angiogenesis analyzer (ImageJ software). Values are summarized as means ± SEM, *P* values were generated using one-way ANOVA with Tukey's post hoc test for comparison. *n* = 3 independent experiments. Source data are provided as a Source Data file.

neovascularization may contribute to disease progression. Intriguingly, in our mouse study we found that *Col15a1* was also enriched in *Ackr1*+ venous ECs, though the expression of this gene was not as distinctive as *Ackr1* and it was also detected in other activated venous ECs. Given that ACKR1+ venules were found to reside in different lung locations and circulatory systems (bronchial and pulmonary circulations), we cannot conclusively establish which venous EC population is involved in the aberrant vascular remodeling associated with persistent fibrosis in aged lungs. Our data show that ACKR1+ veins were also found within the thickened media of arteries in IPF lungs, and thus we speculate that these newly formed vessels may have penetrated the arterial wall causing this vascular abnormality. Notably, vascular malformations involving arteries and veins, such as arteriovenous shunts, have been previously described in IPF lungs[72,73], however, the endothelial origin and the mechanisms implicated in these vascular aberrations have remained largely unexplored.

General capillary gCap ECs have been reported to function as progenitor cells capable of differentiating into aerocytes (aCap ECs) in response to injurious lung stimuli[17]. Our lineage mapping of gCap EC dynamics following bleomycin-induced lung injury in young animals revealed that gCap ECs move to an activated state coinciding with fibrosis onset and that this state transitions back to a healthy gCap EC state with injury resolution. The shift to an activated state was independent of genetics background, as observations in this lineage tracing model were from a mixed C57B/6-129 genetic background mirrored those from C57B/6 injured mice. Our lineage tracing also demonstrated that under these injury conditions gCap ECs do not differentiate into aCap ECs, but rather acquire transient transcriptional changes in young lungs that are sustained in aged lungs. Previous studies have reported that lung capillary ECs can transform into collagen-producing mesenchymal cells (endothelial mesenchymal transition, EndMT) in response to bleomycin-induced lung injury to give rise to pathogenic collagen-producing cells[74]. Our lineage tracing study using genetically labeled capillary ECs demonstrated that although injured capillary ECs transiently overexpressed mesenchymal-associated genes, these cells largely maintained their endothelial identity and do not undergo mesenchymal cell differentiation.

Our data also highlighted the involvement of the mechanosensitive transcriptional regulators YAP and TAZ, which have emerged as key players in fibrotic tissues[75–77]. Pharmacological activation of YAP/TAZ in capillary ECs in vitro promoted the expression of markers associated with an activated/immature endothelial state, resembling the intermediate state of gene expression observed in activated gCap ECs in vivo. Intriguingly, previous studies have shown that activation of YAP/TAZ signaling in various cell types, including those in the lung epithelium, directs cell fate dynamics[78–81]. Taken together with these observations, our findings implicate the YAP/TAZ pathway in response to injury of several lung cell types and suggest

that YAP/TAZ dysfunction may contribute to the progressive fibrosis of aged lungs. Our transcriptional analysis together with lineage tracing identified TrkB as a YAP/TAZ-regulated gene that was distinctly expressed by intermediate activated gCap ECs. TrkB together with its ligand brain-derived neurotrophic factor (BDNF) are best characterized as regulators of neurogenesis[82,83], but they have also been shown to regulate developmental angiogenesis and vessels maturation[46,84,85]. Activation of BDNF-TrkB signaling promotes angiogenesis by influencing vascular endothelial growth factor receptor-2 (VEGFR2) expression[86]. Interestingly, the TrkB ligand BDNF is strongly expressed by ATI cells, which suggests the existence of paracrine crosstalk between epithelial and endothelial cell populations during alveolar repair following injury. Given that our scRNA-seq analysis of the aged lung revealed abnormal ATI transcriptional responses to injury, including reduced expression of *Bdnf* and accumulation of poorly differentiated gCap ECs with aging, these findings suggest that aging-mediated aberrant epithelial cell responses may negatively impact gCap EC regeneration following injury. In support of these findings, our FACS analysis showed that the number of capillary ECs expressing the gCap EC marker TEK were depleted in IPF lungs compared to healthy lungs, further implicating the loss of gCap EC identity as a pathogenic feature of fibrotic lungs with compromised regenerative capacity. Interestingly, we found that BDNF treatment of human lung microvascular ECs in vitro increased the formation of vessel-like structures, and that this was elevated in the presence of TRULI, a small molecule that elevates YAP/TAZ activity and induces the expression of genes associated with the EC "activated" state. These observations suggest a model in which YAP/TAZ activation in gCap ECs following injury promotes an activated state that includes expression of the BDNF receptor TrkB, priming these cells for morphogenesis in response to BDNF, a ligand available from regenerating ATI cells. However, defects in ATI abundance in severely injured areas, such as those that persist in aging, may result in defective EC morphogenesis and an accumulation of an activated EC state, which contributes to fibrogenesis. Thus, restoring regenerative signaling pathways in lung progenitors, including the balance of ATII-ATI cells and signals that promote gCap EC morphogenesis, may provide avenues for enhancing alveolar repair in elderly suffering from chronic lung disorders, including IPF.

Altogether, our study identifies previously unappreciated cellular changes associated with lung injury that vary with age, including putative roles of venous and capillary responses to bleomycin-induced lung injury, and open new research opportunities to explore how lung injury and aging influence endothelial cell behavior in different vascular beds.

## Methods
### Patient samples
Lung tissues from patients with IPF were obtained from explanted lungs obtained at the time of transplantation. All patients provided

written informed consent and the study was approved by the University of Michigan Institutional Review Board, Ann Arbor, MI, USA (HUM00105694). Diagnoses of patients with IPF were established by clinical pathological criteria and confirmed by multidisciplinary consensus conference. Additional IPF lung tissue was obtained from a deceased individual; permission for the autopsy on this case was granted by the next of kin (VA Medical Center, Seattle, WA, USA). Normal control lungs were obtained from deceased donors whose lungs were deemed unsuitable for transplant and were provided by Gift of Life, Michigan, with consent from family for tissue to be used for research purposes. No compensation was provided to subjects or family for either IPF patient samples or normal control lungs. Sex and/ or gender was not considered in the study design. Both sexes were included in the study and distributed randomly in the experiments, and we did not observe any sex-dependent differences in our findings. Due to the limited number of patient samples used in the analysis, statistical adjustment for sex or gender analysis was not performed. In this study, we used lung tissues from five normal donors (two females and three males) and nine patients with IPF (four females and five males).

## Mice

All animal experiments were conducted according to protocols approved by the Institutional Animal Care and Use Committee (IACUC) at Boston University (Protocol # PROTO201900054) and conforming to the Animal Research: Reporting of In Vivo Experiments (ARRIVE) guidelines.

Male young (2 months old) and aged (18 months old) C57BL/6 J mice were purchased from Jackson Laboratory (Bar Harbor, ME, USA). Male and female *Col1a1*-GFP transgenic mice (2 months old, FVB strain) were generated as previously described (UC San Diego, La Jolla, CA, USA)[87] and kindly provided by Dr. Derek Radisky. Male and female (2 months old) Aplnr-Cre-ER(T):Rosa26-mdtTomato/mEGFP reporter mice on a C57/Bl6 background were generated by breeding Aplnr-CreER (Tg(Aplnr-cre/ERT2)[88] (Kindly provided by Dr. Kristy Red-Horse, Stanford University) and B6.129(Cg) – Gt(ROSA)26Sor^tm4(ACTB-tdTomato,-EGFP)Luo^/J (Strain #007676, Jackson Laboratory, Bar Harbor, ME). All mice had access to food and water ad libitum and were on a 12 h/12 h light/ dark cycle, ambient temperature 77–78 °F and humidity 46–49%.

## Bleomycin-induced lung fibrosis model

Bleomycin was delivered to the lungs as previously described[9,89]. Briefly, mice were anesthetized with a ketamine/xylazine solution (100 and 10 mg/kg, respectively) via intraperitoneal injection and administered 1 U/kg of bleomycin (Fresenius Kabi, Lake Zurich, IL, USA) intratracheally (IT). The control group of mice (sham) received PBS IT. Body weight was monitored daily. At the time of lung harvesting, mice were euthanized with 100 µl of FATAL-PLUS solution (Vortech pharmaceuticals, Dearborn, MI, USA).

## Cell culture

Human lung microvascular endothelial cells (HLMECs) were purchased from Cell Applications (540-05a, San Diego, CA, USA) or ANGIO-PROTEOMIE (cAP-0032, Worcester, MA, USA) and maintained in endothelial cell growth basal medium supplemented with microvascular endothelial cell growth kit. Cells were treated with 2 µM of TRULI (E1061, Selleckchem, Houston, TX, USA), a LATS inhibitor and YAP/TAZ signaling activator, for 48 hrs. All the experiments were performed with cells within 3–5 passage.

## Hypoxia detection assay

Hypoxyprobe™-1 RED PE (HP-1 RED PE Mab-1, Hypoxyprobe, Inc; Burlington, MA, USA) was used to characterize hypoxia within the lungs of young and aged mice either sham or injured with bleomycin. 60 days after administration of bleomycin, mice were injected with pimonidazole (60 mg/kg) intraperitoneally 60 min before harvesting of the lungs. For the harvest, mice were euthanized and perfused via left ventricle with cold PBS. Lungs were inflated with a solution of OCT and PBS (50%/50%), followed by embedding of the samples in OCT compound (Tissue-Tek 4583; Sakura Finetek Japan, Co. Ltd, Tokyo, Japan). Tissue sections (5 mm) from each block were cut in a cryostat at -21 C and mounted onto Vectabond-coated slides (Vector Laboratories, Peterborough, UK), fixed in 4% paraformaldehyde (PFA) for 10 minutes at room temperature and blocked in 5% normal donkey serum in 1X PBS containing 0.2% Triton-X. Primary antibodies against pimonidazole (HP-1 RED PE Mab-1, Hypoxyprobe, Inc, 1:100 dilution), anti-CD31 rat antibody (550274, clone MEC 13.3, BD Biosciences, San Jose, CA, USA, 1:200 dilution), Anti-PDPN Syrian hamster antibody (13-5381-82, Clone eBio8.1.1 (8.1.1), Thermo Fisher Scientific, Waltham, MA, USA, 1:300 dilution) and anti-Vimentin rabbit antibody (5741 S, clone D21H3, Cell Signaling Technology, Danvers, MA, USA 1:300 dilution) were incubated in blocking solution overnight at 4 °C. Sections were then incubated for 1 hour with a fluorescence-conjugated secondary antibodies: donkey anti-Rabbit-488(A21206, Thermo Fisher Scientific, 1:1000 dilution), donkey anti-Rat-488 (A21208, Thermo Fisher Scientific, 1:1000 dilution), goat anti-Syrian hamster-488 (107-546-142, Jackson ImmunoResearch, West Grove, PA, USA, 1:1000 dilution) and DAPI (62248, Thermo Fisher Scientific, 1:1000 dilution) to counterstain nuclei. Controls were done by omitting the primary antibodies. Images were captured using a Zeiss Axio Observer.Z1 microscope with Zeiss ZEN 3.3 Blue software, a Zeiss LSM 700 microscope with ZEN 2.3 SP1 FP3 Black software. Two to 6 regions per mouse were analyzed and mean ± SD was determined.

## Single-cell RNA sequencing and data analysis

Young lungs (1 Sham and 1 Bleo-treated) and aged lungs (2 Sham and 2 Bleo-treated) were harvested and dissociated into a single-cell suspension as previously described[8]. Thirty days pos-bleomycin treatment, mice were euthanized and perfused via the left ventricle with 10 mL of sterile cold PBS. The lungs were immediately harvested and minced with a razor blade in a 100 mm petri dish in a cold DMEM medium containing 0.2 mg/ml Liberase DL and 100 U/ml DNase I (Roche, Indianapolis, IN, USA). The mixture was transferred into 15 ml tubes and incubated at 37 °C for 35 min in an incubator under continuous rotation to allow enzymatic digestion. Digestion was inactivated with a DMEM medium containing 10% fetal bovine serum, the cell suspension was passed through a 40 µm cell strainer (Fisher, Waltham, MA, USA) to remove debris. Cells were then centrifuged (500 × g, 10 min, 4 °C), and suspended in 3 ml red blood cell lysis buffer (Biolegend, San Diego, CA, USA) for 90 s to remove the remaining red blood cells and diluted in 9 mL PBS after incubation. Cells were then centrifuged (500 × g, 10 min, 4 °C) and resuspended in 0.2 ml of 1X PBS buffer containing 0.04% BSA. Cell suspensions were submitted to the Single-Cell Sequencing Core (Boston University) for processing. Cell viability and counts were determined using Countess II Automated Cell Counter. Single cells, reagents, and a single Gel Bead containing barcoded oligonucleotides are encapsulated into nanoliter-sized Gel Bead-in Emulsion using the 10x Genomics GemCode platform (10X Genomics, Pleasanton, CA, USA). Lysis and barcoded reverse transcription of RNAs from single cells is performed as described by 10x genomics. Enzyme fragmentation, a tailing, adapter ligation, and PCR are performed to obtain final libraries containing P5 and P7 primers used in Illumina bridge amplification. Size distribution and molarity of resulting cDNA libraries were assessed via Bioanalyzer High Sensitivity DNA Assay (Agilent Technologies, USA). All cDNA libraries were sequenced on an Illumina NextSeq 500 instrument according to Illumina and 10X Genomics guidelines with 1.4–1.8 pM input and 1% PhiX control library spike-in (Illumina, USA). Sequencing data were processed using 10X Genomics' Cell Ranger pipeline to generate feature/ barcode matrices from raw count data.

Feature and barcode matrices of the samples were imported into BioTuring Browser 3 (Bioturing, San Diego, USA) for analysis. After quality filtering, we obtained 52,542 cell profiles, 11767 from young sham, 5893 from young Bleo, 16,507 from aged sham and 18,375 from aged bleo mice. We perform Unsupervised Uniform Manifold Approximation and Projection (UMAP) or t-Distributed Stochastic Neighbor Embedding (t-SNE) dimensionality reduction with canonical correlation analysis (CCA) subspace alignment and performed unsupervised graph-based clustering. Analysis of representative marker genes identified clusters of endothelial, epithelial, and mesenchymal cells as well as hematopoietic cells. Indicated cells were selected and further re-clustered for analysis. The criteria for selection of significant differentially expressed genes were: log2 fold change ≤ −0.1 or ≥0.1 and Benjamini−Hochberg adjusted $P$ value or FDR ≤ 0.05. This list of differentially expressed genes was used for investigating enriched canonical pathways and upstream regulators using Core analysis from the Ingenuity Pathway analysis (IPA, Ingenuity® Systems, www.ingenuity.com). $P$ values were generated in IPA using Fisher's test (Log2 fold change ≤ −0.1 or ≥0.1, $P$ value ≤ 0.05). $P$ value and activation $z$ score were used for plotting noteworthy canonical pathways and activated upstream regulators respectively.

## Temporal lineage tracing scRNA sequencing
Lineage-tracing in Aplnr-Cre-ER(T):Rosa26-mdtTomato/mEGFP reporter mice were used for this experiment. Capillary ECs were permanently labeled using tamoxifen (5 injections of 75 mg tamoxifen/kg body weight, daily). Mice were then injured with bleomycin as described above. Lung tissues from uninjured sham ($n$ = 3), as well as from those harvested at 14 days ($n$ = 2) and 35 days ($n$ = 2) after bleomycin challenge, were collected and the lung tissues dissociated into a single-cell suspension. Briefly, mice were euthanized as described in the method above. After opening the thoracic cavity of mice, the lungs were perfused with 10 mL of sterile cold PBS through the right ventricle of the heart. The exposed trachea was cannulated with a 20 G catheter and injected with 2 mL of digestion solution (PBS containing 80 U/mL collagenase type II (Worthington Biochemical, Lakewood, NJ, USA) and 2 U/mL Dispase II (Sigma-Aldrich), which was followed by clogging with forceps for 2 min. The lung lobes were dissected from the trachea, finely minced, and incubated with 20 mL digestion solution for 1 h in a 37 °C incubator with shaking followed by passaging through an 18 G needle 4–5 times and an additional 15 min incubation at 37 °C. After passing through a 21 G needle, the cell suspension was filtered through 70 and 40 μm mesh cell strainers and centrifuged at 400 × $g$ for 5 min to collect the isolated cells. Red blood cells were lysed using lysis buffer (Sigma-Aldrich). Hematopoietic (CD45$^+$) cells were depleted using magnetic cell separation (MACS). Samples were processed and sequenced, and data were analyzed as described above.

## Quantitative real-time PCR
Total mRNA was isolated using Quick-RNA™ Miniprep (Zymo Research, Irvine, CA, USA) followed by Nanodrop concentration and purity analysis. cDNA was synthesized using High-capacity cDNA Reverse Transcription Kit (Applied Biosystems); RT-PCR was performed using PowerUp™ SYBR™ Green Master Mix (Applied Biosystems) and analyzed using a Step-One-Plus Real-Time PCR system (Applied Biosystems). The PCR primer sequences used in this study are listed in Supplementary Table 1.

## RNA interference
Transient RNA interference was performed with siGENOME non-targeting Control siRNA Pool #1 (D-001206-13-05, 25 nM) or Human YAP/TAZ-siRNA (UGUGGAUGAGAUGGAUACA)[90] by using Lipofectamine RNAiMAX reagent (13778075, Thermo Fisher Scientific, Waltham, MA, USA). Cells were harvested after 72 hrs.

## Immunohistochemistry
Formalin-fixed paraffin-embedded (FFPE) mouse and human lungs were cut in serial sections (7 μm). The FFPE sections were deparaffinized using a standard protocol of xylene and alcohol gradients. Sections were then blocked first with BLOXALL endogenous peroxide blocker (SP-6000-100, Vector Laboratories, Peterborough, UK) and then with 5% goat serum and 2% BSA (Sigma-Aldrich, St. Louis, MA, USA). Staining was performed using the VECTASTAIN Elite ABC HRP kit (PK-6200, Vector Laboratories, Peterborough, UK), anti-CD31 rat antibody (550274, clone MEC 13.3, BD Biosciences, 1:200 dilution), anti- CD31 mouse antibody (131M-94, clone JC70, Cell Marque, Millipore Sigma, USA, 1:200 dilution), anti-TrkB rabbit antibody (4607 S, clone 80G2, Cell Signaling Technology, 1:200 dilution), anti-C1qR1/CD93 goat antibody (AF2379, R&D systems, Minneapolis, MN, USA 1:200 dilution) and the detection with impact DAB (Vector Laboratories, Peterborough, UK). Slides were then dehydrated using a standard protocol and mounted on a coverslip using DPX mountant (Sigma-Aldrich, St. Louis, MA, USA). Masson's trichrome staining was performed by using a commercially available stain kit (HT15, Sigma–Aldrich). Images were captured using an Olympus BH-2 microscope with iVision-Mac 4.5.4 software.

## Immunofluorescence staining
Human or mouse lung tissue slides (7 μm) were fixed in 3.7% formalin (Sigma-Aldrich, St. Louis, MA, USA), permeabilized in 0.1% Triton X-100 (Sigma–Aldrich, St. Louis, MA, USA), blocked with 5% BSA for 1 h. Lung tissue sections were stained with anti-CD31 rabbit antibody (77699, clone D8V9E, Cell Signaling Technology, 1:200 dilution), anti-CAR4 goat antibody (AF2414, R&D systems, 1:200 dilution), anti-TrkB goat antibody (AF1494, Novus, Centennial, CO, USA, 1:200 dilution), anti-αSMA mouse antibody (F3777, clone 1A4, Sigma–Aldrich, 1:200 dilution), anti-mouse ACKR1 rat antibody (kindly provided by Dr. von Andrian, 1:200 dilution), anti-CD31 mouse antibody (3528, clone 89C2, Cell Signaling Technology, 1:200 dilution), anti-ACKR1 goat antibody (NB100-2421, Novus, 1:200 dilution), anti-Col1α1 antibody (72026, clone E8F4L, Cell Signaling Technology, 1:200 dilution),. Sections were stained with fluorescence-conjugated secondary antibodies: donkey anti-Goat-555 (A32816, Thermo Fisher Scientific, 1:1000 dilution), donkey anti-Mouse-488 (A21202, Thermo Fisher Scientific, 1:1000 dilution), donkey anti-Mouse-647 (A31571, Thermo Fisher Scientific, 1:1000 dilution), donkey anti-Rabbit-488 (A21206, Thermo Fisher Scientific, 1:1000 dilution), donkey anti-Rabbit-647 (A31573, Thermo Fisher Scientific, 1:1000 dilution), donkey anti-Rat-488 (A21208, Thermo Fisher Scientific, 1:1000 dilution) and DAPI (62248, Thermo Fisher Scientific, 1:1000 dilution) to counterstain nuclei. Images were captured using an Olympus CKX53 Microscope with INFINITY ANALYZE 6.5.4 software, a Zeiss LSM 700 microscope with ZEN 2.3 SP1 FP3 Black software, or a Zeiss Axio Scan.Z1 microscope with Zeiss ZEN 3.1 Blue software.

## FACS analysis
Human normal and IPF lungs were minced with a razor blade in a 100 mm petri dish in a cold DMEM medium containing 0.2 mg/ml Liberase DL and 100 U/ml DNase I (Roche, Indianapolis, IN, USA). The mixture was transferred into 15 ml tubes and incubated at 37 °C for 35 min in a water bath under continuous rotation to allow enzymatic digestion. Digestion was inactivated with a DMEM medium containing 10% fetal bovine serum, the cell suspension was passed through a 40 μm cell strainer (Fisher, Waltham, MA, USA) to remove debris. Cells were then centrifuged (500 × $g$, 10 min, 4 °C), and resuspended in 3 ml red blood cell lysis buffer (Biolegend) for 90 s to remove the remaining red blood cells and diluted in 9 mL PBS after incubation. Cells were then centrifuged (500 × $g$, 10 min, 4 °C) and resuspended in 0.2 ml of FACS buffer (0.05% BSA, 0.5 mM EDTA pH 7.4 in PBS). Single cell

suspension was stained with antiCD45:Pacific blue (368539, clone 2D1, Biolegend, 1:200 dilution), anti-EpCAM:BV650 (324225, colon 9C4, Biolegend, 1:200 dilution), anti-CD31:APC/Cyanine7 (303119, clone WM59, Biolegend, 1:200 dilution), anti-Tek:PE (334205, clone 33.1, Biolegend, 1:100 dilution), anti-ACKR1 (NB100-2421, Novus, 1:50 dilution), anti-P-Selectin (NB100-65392, Novus, 1:50 dilution), anti-CTHRC1 (PA5-49638, Invitrogen, Waltham, MA, USA, 1:50 dilution), anti-rabbit IgG:FITC (406403, clone Poly4064, Biolegend, 1:200 dilution), anti-goat IgG:PE (405307, clone Poly4053, Biolegend, 1:200 dilution) and DAPI (D3571, Thermo Fisher Scientific, 1:10,000 dilution) for 15 min on ice. Cells were washed with ice-cold FACS buffer twice and analyzed by BD LSR II (BD Biosciences). FACS analysis was performed with the following strategy: debris exclusion (FSC-A by SSC-A), doublet exclusion (SSC-W by SSC-H and FSC-W by FSC-H), and dead cell exclusion (DAPI by FSC-A). Data were analyzed with FlowJo version 10.8.0 software (Tree Star Inc., Ashland, OR, USA).

### In vitro 3D endothelial morphogenesis assay

Capillary morphogenesis assay was performed as previously described[91]. HLMECs ($2 \times 10^6$ cells/ml) were embedded into collagen type-I gel (2 mg/ml) containing 1 μM Sphingosine 1-phosphate. 10 μl of the collagen/cell mixture was disposed into μ-Slide wells (Ibidi, Munich, Germany) and allowed to gel at 37 °C and 5% $CO_2$ for 30 min. Cultures were kept in Endothelial Cell Growth Medium MV2 (PromoCell, Heidelberg, Germany) containing 50 ng/ml VEGF and supplied with DMSO, BDNF (50 ng/ml) or TRULI (2 μM) for 48 h. Cultures were fixed in 4% PFA for 1 h at room temperature and stained with Phalloidin-iFlor 555 (Abcam, Cambridge, MA, USA). Capillary networks were quantified using angiogenesis analyzer (ImageJ software).

### Statistics and reproducibility

Individual data points are shown in all plots and represent data from independent mice, cells, or biological replicates from cell culture experiments. All bar graphs indicate mean ± SEM, with statistical analysis performed using Unpaired two-tailed Student's $t$-test or one-way ANOVA with Tukey's post hoc test for comparison. Boxes in all box plots extend from the 25th to 75th percentiles, with a central line at the median, while whiskers show minimum and maximum values unless specified otherwise in figure legends. The number of animals, cells and experiments in each group are indicated in the figure legends, with $P$ values for comparisons shown in the figures. Randomization applies to all statistical analyses and the allocation of mice to treatment groups. Data collection and analysis procedures were not conducted blind to the experiment operators. Analysis of publicly available scRNA-seq dataset from human IPF and healthy lungs (GSE136831) was performed using R packages, version 4.0.3. All analyses, plots, and heatmaps were generated using GraphPad Prism 9.4.1 or BioTuring Browser 3, with statistical significance defined as $P < 0.05$. For analyses by $t$-test and ANOVA, the GraphPad Prism program provides exact $P$ values for $P = 0.0001$ or higher; for lower values, it yields $P < 0.0001$. Venn diagrams were created using Venny 2.0 (https://bioinfogp.cnb.csic.es/tools/venny/index2.0.2.html).

### Reporting summary

Further information on research design is available in the Nature Portfolio Reporting Summary linked to this article.

## Data availability

The scRNA-seq data generated in this study have been deposited in the Gene Expression Omnibus under the GEO accession numbers GSE264151 and GSE264162. The publicly available scRNA-seq dataset from human IPF and healthy lungs used in this study are available in the GEO database under accession code GSE136831. All the remaining data are available within the Article, Supplementary Information or Source Data file. Source data are provided with this paper.

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

## Acknowledgements

We would like to acknowledge support from the Boston University Flow Cytometry Core and Microarray and Sequencing Resource Core Facility, with particular thanks to Yuriy Alekseyev, Tianmu Hu and Salam AlAbdullatif. We thank Dr. David Rogers, Pathology and Laboratory Medicine, VA Puget Sound Health Care System, Seattle, WA for his help in providing histologic sections of postmortem lung tissue. We also thank Dr. Kristy Red-Horse (Department of Biology, Stanford University) for sharing the Aplnr-CreER(T) mouse line. We gratefully acknowledge support of this work by the National Institutes of Health (NIH) grants R01HL142596 (G.L.), R01HL158733 (G.L.), R01HL124392 (X.V.), T32HL007035 (T.X.P.) and support from The Evans Center for Interdisciplinary Biomedical Research ARC on "Connecting Tissues and Investigators, Fibrosis in Pathology" at Boston University.

## Author contributions

G.L. and X.V. conceptualized the study and acquired funding. A.A.R., T.X.P., J.L., K.K., A.T-L., J.S., J.H., and N.C. performed the experiments. A.A.R., T.X.P., G.L., X.V., and T.D. analyzed and visualized the data. A.T. and U.H.V.A. developed the antibody against mouse ACKR1. The manuscript was drafted by G.L. and A.A.R., and revised by X.V., M.T., R.F.N., A.M.B., S.K.H., N.C., and U.H.V.A. Human lung samples were procured by S.K.H. and R.F.N. All Authors participated in manuscript preparation and provided final approval of the submitted work.

## Competing interests

The authors declare no competing interests.
