## [Peer Review File · Nature Communications]

Reviewers' comments:

Reviewer #1 (Remarks to the Author):

This data is a preliminary study that implies interesting observations but as it is based on n=1 in the young mice and n=2 in the aged mice it is hard to assess the significance of these studies.

Reviewer #2 (Remarks to the Author):

The study by Raslan et al. investigates age-dependent changes in lung injury and regeneration caused by bleomycin injection in mice. They identified *Pdgfra* + alveolar fibroblasts as a major source of collagen expression following bleomycin challenge, which was more pronounced in aged lungs. Age-associated transcriptional abnormalities were observed that potentially affected lung progenitor cells, including ATII pneumocytes and general capillary (gCap) endothelial cells (ECs). Transcriptional analysis combined with lineage tracing identified a sub-population of gCap ECs marked by the expression of Tropomyosin Receptor Kinase B (TrkB) that appeared in bleomycin-injured lungs and accumulated with aging. The TrkB+ EC population expressed common gCap EC markers but also exhibited a distinct gene expression signature associated with aberrant YAP/TAZ signaling, mitochondrial dysfunction, and hypoxia. Finally, the authors defined ACKR1+ venous ECs that exclusively emerged in injured lungs of aged animals and were closely associated with areas of collagen deposition and inflammation. Elegant lineage tracing experiments as well as comprehensive single cell phenotyping analyses were carefully performed. However, several concerns were raised that require clarification.

1. Issues with statistical significance – key data on single cell analysis from n=1 or n=2 for control and bleomycin-treated groups. Given the significant individual variance of mice response to bleomycin, more mice need to be included to each group.
2. The study has been performed in male mice. This limitation must be acknowledged, and discussion of results must be provided in this context
3. Correlations between cultured HLMVEC and capillary EC in situ are not convincing. Unclear if this feature is specific for capillary EC sub-phenotype or for all cultured EC
4. Background: original study (Angelidis et al., Nat Communications, 2019) that presented a first single cell atlas of aging lungs and identified sub-phenotypes with age-dependent alterations of transcriptional program has been ignored by the authors
5. Studies by other groups identified 30 and 40 distinct lung cell types compared to 15 cell types reported in this study. These discrepancies must be discussed.

6. Introduction is written in non-conventional manner: it summarizes results of this study rather than provides more comprehensive review of single cell phenotyping studies performed by other groups.
7. Discussion is overly speculative, and some conclusions are not supported by mechanistic verification studies.
8. Overall, the study is confirmatory of previous reports in the literature and provides incremented information.

Reviewer #3 (Remarks to the Author):

The current manuscript by Raslan et al used single-cell transcriptomics to profile lung cells in fibrosis during tissue aging in mice. They compared the scRNAseq data between young and aged mice lungs with/without the treatment with bleomycin. They determined that fibroblasts display persistent activation during aging compared to their counterparts in young animals. Aging also reduces the regenerative potential in ATII and gCap cells, especially after bleo treatment. They also found that gCap activation may be regulated through YAP/TAZ and NTRK2-dependent mechanisms. Overall, this is an interesting and well-written manuscript. However, the rigor is largely limited by their small sample size and lack of confirmatory experiments.

1. Given the experimental conditions (young vs. aged; sham vs. bleo) the author is testing, the sample size is too small to make reliable conclusions.
2. Confirmatory experiments such as immunostaining are generally lacking in young, old, sham, and bleo lungs throughout the entire manuscript, which reduces the rigor of the current manuscript.
3. In Figure 3, authors should consider adding computational analysis (velocity or pseudotime) to predict cellular transition in the intAE cluster between aged and young cells.
4. TEK is generally considered a pan-endothelial cell marker. Although previous literature shows the expression of this gene is slightly higher in gCap cells compared to other ECs, it is hardly used as a marker to distinguish gCap from other lung ECs, especially in FACS and immunostaining assays. A more specific marker should be used, such as Aplnr.
5. Figure 5 shows the possible relationship between TrkB expression and aging and lung injury. However, the paper did not provide any direct evidence confirming the regulation other than an in vitro HLMVEC study. A lineage tracing study would help to determine the direct interaction.
6. The featurePlot in figure 6 shows that cluster 3 expressed lower aCap markers (Kdr, Ednrb, Car4) and higher gCap markers (Kit, Plvap) compared to clusters 1, 2. How do you rule out the possibility that cluster 3 is NOT aCap cells?
7. The lung contains many different cell populations that work jointly to maintain its function and homeostasis. The paper showed significant change across multiple individual cell populations. It would be interesting to see the changes in cell-cell communications during different conditions.

Response to reviewers: Manuscript NCOMMS-23-05914

We thank the reviewers for their effort in evaluating our manuscript, their constructive comments, and the recognition of our work.

Reviewers' comments:

Reviewer #1 (Remarks to the Author):

This data is a preliminary study that implies interesting observations but as it is based on n=1 in the young mice and n=2 in the aged mice it is hard to assess the significance of these studies.

We agree that number of mice analyzed by scRNA-seq in the original manuscript (aging dataset) were limited, but this analysis was carried out as an initial screen to gain insights into aging-associated lung cellular and transcriptional responses to injury/fibrosis. To further validate our findings and add important conceptual insights into vascular responses to lung injury we have carried out an additional scRNA-seq study using a newly developed reporter mice in which capillary lung endothelial cells (ECs) are genetically labeled. For this study we analyzed lung vascular response during the acute phase post bleomycin injury (day 14) and the initial resolving phase (day 30). Using this approach we were able to assess the fate and temporal transcriptional changes of pulmonary endothelial cells at single cell resolution. This new analysis not only confirms the key findings we report in the original scRNA-seq dataset (young vs aged) but also revealed, for the first time, the fate and phenotypic transition of lung capillary EC post injury and during lung fibrosis resolution.

In addition, we have provided additional data validating our major findings reporter in the initial version of the manuscript, including a lineage tracing study demonstrating the acquisition of a transient activated state in capillary ECs marked by the expression of TrkB. We have also showed by immunohistochemistry that TrkB positive ECs are topologically restricted to areas of lung remodeling and accumulate in aged lungs in which fibrosis is more persistent. We also validated these important findings in human lungs from IPF patients. We feel that our revised manuscript not only validate our initial transcriptional analysis, but also provide a previously undescribed mechanism of capillary injury in response to lung damage. These mechanisms include an activation of gCap ECs following bleomycin-induced lung injury that is transient in young mice but persists in aged mice. This fate of gCap ECs differs from that described with other lung injury stimulus, such as elastase-induced injury during which gCap EC undergo differentiation into specialized aCap EC (Ref).

Another new discovery we made using independent scRNA-seq datasets was the emergence of a population of vein endothelial cells marked by ACKR1 following bleomycin-induced injury. These findings were validated by immunohistochemistry in mouse and human lung samples using immunofluorescence and FACS analysis. Intriguingly ACKR1+ ECs have also been identified in the liver of patients with chronic liver disease (PMID: 31597160) supporting our findings and suggesting an underappreciated mechanism that may span different organs.

We did recognize the need of additional experiments to strengthen our conclusions and therefore we performed several more experiments to support our conclusions, including an additional scRNA-seq study. We also restructured the presentation of our data to better highlight the key conclusions and limit the presentation of our data as a "cell atlas". We hope the that the reviewer agrees that our revised manuscript offers novel and insightful data that will be important for the field of lung injury.

Reviewer #2 (Remarks to the Author):

The study by Raslan et al. investigates age-dependent changes in lung injury and regeneration caused by bleomycin injection in mice. They identified Pdgfra + alveolar fibroblasts as a major source of collagen expression following bleomycin challenge, which was more pronounced in aged lungs. Age-associated transcriptional abnormalities were observed that potentially affected lung progenitor cells, including ATII pneumocytes and general capillary (gCap) endothelial cells (ECs). Transcriptional analysis combined with lineage tracing identified a sub-population of gCap ECs marked by the expression of Tropomyosin Receptor Kinase B (TrkB) that appeared in bleomycin-injured lungs and accumulated with aging. The TrkB+ EC population expressed common gCap EC markers but also exhibited a distinct gene expression signature associated with aberrant YAP/TAZ signaling, mitochondrial dysfunction, and hypoxia. Finally, the authors defined ACKR1+ venous ECs that exclusively emerged in injured lungs of aged animals and were closely associated with areas of collagen deposition and inflammation. Elegant lineage tracing experiments as well as comprehensive single cell phenotyping analyses were carefully performed. However, several concerns were raised that require clarification.

1. Issues with statistical significance – key data on single cell analysis from n=1 or n=2 for control and bleomycin-treated groups. Given the significant individual variance of mice response to bleomycin, more mice need to be included to each group.

We agree that number of mice was a limitation of our initial study and thus to further validate our findings and add important conceptual insights into cellular and transcriptional responses to lung injury we have carried out a temporal scRNA-seq experiment using a newly developed reporter mouse. In addition, we have provided additional data validating our major findings, including immunostaining and flow cytometry analysis of mouse and human fibrotic tissues, along with analyses of scRNA-seq of human lungs with pulmonary fibrosis.

2. The study has been performed in male mice. This limitation must be acknowledged, and discussion of results must be provided in this context.

We agree that use of one sex is a limitation and is something we will point out in the discussion of a revised manuscript. However, we note that human fibrosis is predominantly associated with males (~70% of IPF, PMID: 28232409, 18321929, 31771560), which was the rationale for using male animals in our study.

3. Correlations between cultured HLMVEC and capillary EC in situ are not convincing. Unclear if this feature is specific for capillary EC sub-phenotype or for all cultured EC.

We used commercially available lung microvascular endothelial cells (HLMVEC) as a model to show that activation of YAP recapitulates transcriptional features associated with lung capillary EC activation in mice. In our revised version we also included a new vascular morphogenic assay to show the synergistic function of YAP and TrkB signaling pathways. We recognize that commercially available lung capillary EC may not behave as those in vivo, however, we also believe that our data represent an important proof-of concept of the synergistic function of injury-and cellular-associated signaling pathways to lung vascular repair.

4. Background: original study (Angelidis et al., Nat Communications, 2019) that presented a first single

cell atlas of aging lungs and identified sub-phenotypes with age-dependent alterations of transcriptional program has been ignored by the authors

We thank the reviewer for pointing this out and we have included this study in our manuscript.

5. Studies by other groups identified 30 and 40 distinct lung cell types compared to 15 cell types reported in this study. These discrepancies must be discussed.

We are aware that previous studies have reported higher numbers of lung cell types in scRNAseq studies. These numbers could be recapitulated in our study by varying the stringency of cluster analysis, but for simplicity of presentation we chose to show our whole lung cluster analysis, now displayed in Supplementary Fig.1, based on pan-marker genes defining key cell types, such as epithelial cells (EpCam+), fibroblasts (Col1a1+), Endothelial cells (Cd31+), etc., and thus we did not highlight specific cellular subtypes. However, our analysis of key cell types later in the manuscript we have re-clustered cell populations of interest, which highlight additional cell subtypes.

6. Introduction is written in non-conventional manner: it summarizes results of this study rather than provides more comprehensive review of single cell phenotyping studies performed by other groups.

We thank the reviewer for this comment and as suggested we have edited the introduction

7. Discussion is overly speculative, and some conclusions are not supported by mechanistic verification studies.

We have edited the discussion to focus more on key points that are supported by our data

8. Overall, the study is confirmatory of previous reports in the literature and provides incremented information.

We respectfully disagree with the reviewer on this point. While there are previous scRNA-seq studies that have been performed, there are several new data and conclusions provided by the observations in our manuscript, including the identification of new injury associated states of endothelial cells (veins and capillary), and new pathways involved in endothelial and epithelial cell activation in response to lung injury and in aging. In our revised manuscript we have now included a temporal lineage tracing studies at single cell resolution that provides novel and previously unappreciated insights into capillary responses to lung injury. We would argue that our findings also represent a paradigm-shift in our understanding of vascular responses to bleomycin injury and revealed, for the first time, novel signaling pathways governing capillary EC responses.

Reviewer #3 (Remarks to the Author):

The current manuscript by Raslan et al used single-cell transcriptomics to profile lung cells in fibrosis during tissue aging in mice. They compared the scRNAseq data between young and aged mice lungs with/without the treatment with bleomycin. They determined that fibroblasts display persistent

activation during aging compared to their counterparts in young animals. Aging also reduces the regenerative potential in ATII and gCap cells, especially after bleo treatment. They also found that gCap activation may be regulated through YAP/TAZ and NTRK2-dependent mechanisms. Overall, this is an interesting and well-written manuscript. However, the rigor is largely limited by their small sample size and lack of confirmatory experiments.

1. Given the experimental conditions (young vs. aged; sham vs. bleo) the author is testing, the sample size is too small to make reliable conclusions.

In our revised manuscript we have included a new scRNA-seq study to validate our conclusions and provide additional insights into lung vascular responses to injury. Furthermore, we have included a new analysis using publicly available scRNA-seq studies of IPF lungs to validate our findings in mice. Finally, we have complemented our transcriptional analysis with additional immunostaining, lineage tracing studies, as well as provided functional data using vascular morphogenic assay ex vivo.

2. Confirmatory experiments such as immunostaining are generally lacking in young, old, sham, and bleo lungs throughout the entire manuscript, which reduces the rigor of the current manuscript.

We have added confirmatory experiments using immunostaining analysis as described above.

3. In Figure 3, authors should consider adding computational analysis (velocity or pseudotime) to predict cellular transition in the intAE cluster between aged and young cells.

We have restructured the presentation of our revised manuscript to focus primarily on EC alterations that occur following bleomycin-induced injury and have limited our analysis of the epithelium given the characterization of these cells has been the focus of prior studies. We note that the cellular transition of the intAE cluster using pseudotime analysis has been performed previously and our observations are consistent with these prior reports.

4. TEK is generally considered a pan-endothelial cell marker. Although previous literature shows the expression of this gene is slightly higher in gCap cells compared to other ECs, it is hardly used as a marker to distinguish gCap from other lung ECs, especially in FACS and immunostaining assays. A more specific marker should be used, such as Aplnr.

We agreed with the reviewer that TEK is also expressed in endothelial cells from other vascular beds, however, we and other have previously shown that TEK is highly expressed in gCap ECs relative to other lung ECs. Indeed, FACS analysis can clearly distinguish these cell type versus other ECs based on TEK high expression. We did several attempts using commercially available antibodies against APLNR, but none of the antibodies we have tried were high enough quality for robust FACS or immunostaining-related applications.

5. Figure 5 shows the possible relationship between TrkB expression and aging and lung injury. However, the paper did not provide any direct evidence confirming the regulation other than an in vitro HLMVEC study. A lineage tracing study would help to determine the direct interaction.

In our revised manuscript we carried out a new scRNA-seq study using our gCap EC report mouse and demonstrated that TrkB is transiently but specifically expressed in injured capillary ECs following bleomycin-induced lung injury. We have also included new mouse and human data showing that the TrkB ligand BDNF is almost exclusively expressed by ATI cells and its expression is lost in lungs with sustained fibrosis. We believe that the intimate connection between ATI cells and gCap ECs provides a functional niche that responds to alveolar injury and BDNF/TrkB signaling pathway is critical for these reparative/fibrotic responses.

We used commercially available lung microvascular endothelial cells (HLMVEC) as a model to show that activation of YAP recapitulates transcriptional features associated with lung capillary EC activation in mice. To strengthen our findings we have now provided a morphogenic vascular assay to further validate the synergism between YAP and TrkB signaling pathways.

6. The featurePlot in figure 6 shows that cluster 3 expressed lower aCap markers (Kdr, Ednrb, Car4) and higher gCap markers (Kit, Plvap) compared to clusters 1, 2. How do you rule out the possibility that cluster 3 is NOT aCap cells?

In our revised manuscript we have substantially modified this figure. We have now performed a combined analysis of gCap and aCap EC. In addition, we have carried out a new lineage tracing RNA-seq studies that provides additional novel insights into capillary EC differentiation following bleomycin injury.

7. The lung contains many different cell populations that work jointly to maintain its function and homeostasis. The paper showed significant change across multiple individual cell populations. It would be interesting to see the changes in cell-cell communications during different conditions.

In the previous version of our manuscript we only focused on the emergence of different populations of lung vascular endothelial cells and less on cell-cell communication. In our revised manuscript, however, we provide a new transcriptional analysis focused on alveolar epithelial cells which showed that ATI cells are the main source of the TrkB ligand BDNF and suggest that gCap EC/ATI cell communication is critical to alveolar repair in response to injury.

REVIEWER COMMENTS

Reviewer #1 (Remarks to the Author):

This work by Raslan et al. demonstrates with cRNA-seq analysis revealed numerous aging-associated cellular and transcriptional alterations in response to bleomycin injury that affected many cell populations, including ECs, which exhibited a shifted molecular state that we labelled as “activated” with this state persisting in fibrotic aged and IPF lungs.

The previous submission was largely criticized for the limited sample size of n=1-2. Authors have now included an additional dataset of young mice in a slightly different genetic background C57/Bl6 vs Bl6.129, differences in bleo induced fibrosis activation was not considered. In figure 1 F and O expression of LRG1 and Fxyd5 is different between young mice of the two different experiments. Some of the activation markers have now been validated to be associated with areas of fibrosis although the sample size again is on the low end with 2-3 animals pre group and 2-6 regions per animal. Also, some of the histology and FACS validation were performed on post bleo 28 and 37 whereas the scRNA data was from post bleo 35.

Increase of hyperoxia in EC was validated by hypoxia assessment in sections. Pecan negative areas correlated with high Pimonidazole activity. What cells are in these areas remains unclear.

The following statements are correlative and speculative 318-321, 360-369, 368-369, 387, 409-410. In line 348 they claim endothelial cells were newly formed, what if all the epithelial cells died off and the remaining structure is fibrotic with venous vascularization.

Fig 3 E colocalization is hard to see. The FACS data relies on Thy1 and CTHRC1 which is a very simplistic view of fibroblasts. Digest of tissue with different degree of fibrosis might very likely result in different cell recovery, which is not representative of the starting material but an artefact of the excess of extra cellular matrix.

483-530 is taken out of context and does not add to the story but shows what is known, namely that suppression of LATs in the presence of BDNF induce vascular morphogenesis but how this would be linked to activated vascular cells from bleo and aged remains unclear.

What the author show is that Ackr1 is a nice venous endothelial marker, so is Trkb for activated endothelial cells. The authors showed nicely that Aplnr lineage traced cells do not transdifferentiate into mesenchymal cells. The authors missed the opportunity to interrogate the crosstalk of ACKR1 endothelial cells with fibroblasts. Similar is Trkb causing fibrosis or is fibrosis activating Trkb expression.

Reviewer #2 (Remarks to the Author):

The authors have been responsive to previous review and addressed many points. However, this reviewer recommends to include number of experiments and/or analyzed single cells in the figure legends to each panel.

Response to reviewers: Manuscript NCOMMS-23-05914A-Z

We express our gratitude to the reviewers for dedicating their time to evaluate our manuscript. We appreciate their constructive comments and recognition of our work. We have outlined responses (outlined in blue) to the comments raised below.

Reviewers' comments:

Reviewer #1 (Remarks to the Author):

Comment:

This work by Raslan et al. demonstrates with cRNA-seq analysis revealed numerous aging-associated cellular and transcriptional alterations in response to bleomycin injury that affected many cell populations, including ECs, which exhibited a shifted molecular state that we labelled as “activated” with this state persisting in fibrotic aged and IPF lungs.

The previous submission was largely criticized for the limited sample size of n=1-2. Authors have now included an additional dataset of young mice in a slightly different genetic background C57/Bl6 vs Bl6.129, differences in bleo induced fibrosis activation was not considered.

Response:

We acknowledged the use of slightly different genetic backgrounds in our experiments. However, we do not view this as a limitation of our study, but rather as a strength, as phenotypes overserved are consistent across models. Our observations indicate similar transcriptional and cellular responses across the different mouse strains, including the de novo TrkB expression in gCap endothelial cells induced by bleomycin injury. This consistency strengthens our findings and demonstrates that our results are not only relevant to a specific genetic background.

Comment:

In figure 1 F and O expression of LRG1 and Fxyd5 is different between young mice of the two different experiments.

Response:

The different gene expression is due to the fact that the heatmaps were generated from two independent datasets with distinct sequencing depths. However, we do recognize that this may generate confusion for the readers and thus to enhance clarity, we have replaced the heatmap in Figure 10 with a Venn diagram outlining the overlap in genes. We have also included a IPA pathway analysis in supplemental data to better display transcriptional similarities between the acute phase post bleomycin (day 14) and the resolution phase in young and aged mice ECs.

Comment:

Some of the activation markers have now been validated to be associated with areas of fibrosis although the sample size again is on the low end with 2-3 animals pre group and 2-6 regions per animal. Also, some of the histology and FACS validation were performed on post bleo 28 and 37 whereas the scRNA data was from post bleo 35.

Response:

We agree that some of the chosen timepoints were different among experiments. Based on our previous published data (PMID: 32691484), we consider >30 days a resolving phase of bleomycin injury. However, to consider potential differences in injury responses, we chose to harvest lungs 5- to 7-days after the 30 days timepoint. The 28 days timepoint was chosen to demonstrate the acquisition of TrkB expression by gCap ECs after bleomycin injury.

Comment:

Increase of hyperoxia in EC was validated by hypoxia assessment in sections. Pecan negative areas correlated with high Pimonidazole activity. What cells are in these areas remains unclear.

Response:

We have performed additional experiments and showed that hypoxic signal was mainly detected in PDPN+ type I epithelial cells (ATI) as well as in vimentin positive mesenchymal cells. These data were included in Supplementary Fig.5.

Comment:

The following statements are correlative and speculative: 318-321:360-369: 368-369, 387, 409-410.

Response:

We thank the reviewer for pointing out the correlative nature of some of our comments. We have revised the text and moved some notional concepts to the discussion.

Comment:

In line 348 they claim endothelial cells were newly formed, what if all the epithelial cells died off and the remaining structure is fibrotic with venous vascularization.

Response:

In normal lungs, ACKR1 is primarily expressed in bronchial venous endothelial cells and less in the alveolar region. However, in IPF lungs ACKR1+ ECs were also found in fibrotic areas of the distal lungs, suggesting that venous vascular remodeling occurs in the lung of these patients.

Comment:

Fig 3 E colocalization is hard to see.

Response:

We agree with the reviewer and have changed the color scheme and the overall appearance of the figure have been modified to enhance co-localization and improve visual representation.

Comment:

The FACS data relies on Thy1 and CTHRC1 which is a very simplistic view of fibroblasts. Digest of tissue with different degree of fibrosis might very likely result in different cell recovery, which is not representative of the starting material but an artefact of the excess of extra cellular matrix.

Response:

We used Thy1 as a pan-marker of fibroblasts. This marker allowed us to first distinguish lung fibroblasts from the rest of the lung cells. We also use an antibody against CTHRC1 as this marker was shown to be distinctly expressed in activated collagen-producing cells in mouse and human fibrotic lungs (PMID: 32317643). In our study, we employed CTHRC1 as a marker to address the varying degrees of fibrosis in the tissue. In addition, our immunostaining analyses shown in Fig.3, confirmed the presence of ACKR1 venous ECs in fibrotic areas.

Comment:

483-530 is taken out of context and does not add to the story but shows what is known, namely that suppression of LATs in the presence of BDNF induce vascular morphogenesis but how this would be linked to activated vascular cells from bleo and aged remains unclear.

Response:

In our IPA analysis of upstream regulators, we identified YAP1 as putative transcriptional regulator of EC activated state in response to lung injury (Fig.1E). Our *in vitro* data also showed that TRULI-mediated YAP activation in human lung capillary ECs mirrored the transcriptional signature of activated mouse lung EC *in vivo*, implicating a role for YAP in capillary EC transcriptional reprogramming after lung injury. In addition, the synergistic function between YAP and TrkB signaling pathways in promoting vascular morphogenesis *in vitro* suggest that the activation of both pathways is necessary to restore normal capillary homeostasis following injury. Our data reporting reduced BDNF expression in aged mouse and human fibrotic lungs support the notion that limited TrkB signaling in capillary ECs results in aberrant vascular/angiogenic responses following lung injury and perpetuate fibrosis. We have revised this section in the results and added additional discussion to better clarify these conclusions.

Comment:

What the author show is that Acker1 is a nice venous endothelial marker, so is Trkb for activated endothelial cells.

Response:

ACKR1 is exclusively expressed in venous ECs and its expression increases during fibrotic peak. Our data demonstrate that TrkB is only expressed in activated capillary ECs following injury and it is not expressed in ACKR1+ venous ECs.

Comment:

The authors showed nicely that Aplnr lineage traced cells do not transdifferentiate into mesenchymal cells. The authors missed the opportunity to interrogate the crosstalk of ACKR1 endothelial cells with fibroblasts. Similar is Trkb causing fibrosis or is fibrosis activating Trkb expression.

Response:

We agree with the reviewer that understanding how the crosstalk between ACKR1 and TrkB-expressing endothelial cells with fibroblasts (and other cells) contributes to fibrosis is an important next step. Our discussion speculates on potential mechanisms by which the ACKR1 and TrkB-expressing EC states contribute to defects in fibrosis resolution. While our efforts to gain such knowledge is ongoing, our manuscript has extensive data with new insights into fibrosis development in the lung that should be shared with the scientific community.

Reviewer #2 (Remarks to the Author):**Comment:**

The authors have been responsive to previous review and addressed many points. However, this reviewer recommends to include number of experiments and/or analyzed single cells in the figure legends to each panel.

Response:

We thank the reviewer for the positive response. We have revised the figure legends according to the suggestions provided by the reviewer.

REVIEWERS' COMMENTS

Reviewer #1 (Remarks to the Author):

The authors made an effort to address my earlier concerns regarding rigor, yet they overlooked the issue of inadequate biological replicates and controls for mouse backgrounds. Consequently, certain conclusions still appear speculative, relying solely on circumstantial evidence.

Response to reviewers: Manuscript NCOMMS-23-05914B

We express our gratitude to the reviewers for dedicating their time to evaluate our manuscript. We appreciate their constructive comments and recognition of our work. We have outlined responses (outlined in blue) to the comment raised below.

Reviewers' comments:

Reviewer #1 (Remarks to the Author):

Comment:

The authors made an effort to address my earlier concerns regarding rigor, yet they overlooked the issue of inadequate biological replicates and controls for mouse backgrounds. Consequently, certain conclusions still appear speculative, relying solely on circumstantial evidence.

Response:

We thank the reviewer for recognizing our effort in addressing previous concerns. We added in the discussion the limitation of our work, including number of biological replicates and different mouse backgrounds.